# Dendritic growth and synaptic organization from activity-independent cues and local activity-dependent plasticity

Jan H Kirchner[1,2†], Lucas Euler[2†], Ingo Fritz[1], André Ferreira Castro[1], Julijana Gjorgjieva[1,2*]

[1]School of Life Sciences, Technical University of Munich, Freising, Germany; [2]Computation in Neural Circuits Group, Max Planck Institute for Brain Research, Frankfurt, Germany

## eLife Assessment

This **important** work investigates how two distinct processes, morphological changes and synaptic plasticity, contribute to the final shape of neuronal dendrites and the spatial structure of their synaptic inputs. The modelling is **convincing** and could be broadly applied to other similar questions. The work will be of interest to neuroscientists studying dendritic development and connectivity at a single-cell level.

*For correspondence:
gjorgjieva@tum.de

†These authors contributed equally to this work

**Abstract** Dendritic branching and synaptic organization shape single-neuron and network computations. How they emerge simultaneously during brain development as neurons become integrated into functional networks is still not mechanistically understood. Here, we propose a mechanistic model in which dendrite growth and the organization of synapses arise from the interaction of activity-independent cues from potential synaptic partners and local activity-dependent synaptic plasticity. Consistent with experiments, three phases of dendritic growth – overshoot, pruning, and stabilization – emerge naturally in the model. The model generates stellate-like dendritic morphologies that capture several morphological features of biological neurons under normal and perturbed learning rules, reflecting biological variability. Model-generated dendrites have approximately optimal wiring length consistent with experimental measurements. In addition to establishing dendritic morphologies, activity-dependent plasticity rules organize synapses into spatial clusters according to the correlated activity they experience. We demonstrate that a trade-off between activity-dependent and -independent factors influences dendritic growth and synaptic location throughout development, suggesting that early developmental variability can affect mature morphology and synaptic function. Therefore, a single mechanistic model can capture dendritic growth and account for the synaptic organization of correlated inputs during development. Our work suggests concrete mechanistic components underlying the emergence of dendritic morphologies and synaptic formation and removal in function and dysfunction, and provides experimentally testable predictions for the role of individual components.

## Introduction

The dendrites of a neuron are intricately branched structures that receive electrochemical stimulation from other neurons. The morphology of dendrites determines the location of synaptic contacts with

other neurons, thereby constraining single-neuron computations. During development, the dendrites of many neurons grow simultaneously and become integrated into neural circuits. The development of dendrites is highly dynamic; iterated addition and retraction of branches allow these dendrites to probe various potential synaptic partners before stabilizing (*Cline, 2016*; *Richards et al., 2020*). Many intrinsic and extrinsic factors underlie the dynamics of dendritic development. In any given neuron, the intrinsic expression of specific genes controls many morphological aspects, including the orientation of the dendrite in the cortex, the general abundance of dendritic branching, and the timing of growth onset (*Puram and Bonni, 2013*). In contrast, extrinsic signaling exerts precise control over the detailed dynamics of dendrite development through various mechanisms, including activity-dependent cell-to-cell interactions and molecular signaling (*Polleux et al., 2016*).

Although many signaling molecules affect dendrite development, brain-derived neurotrophic factor (BDNF) and its immature predecessor proBDNF are particularly crucial in the central nervous system (*Lu et al., 2005*). While exposure to BDNF leads to larger dendrites with a higher density of synapses (*McAllister et al., 1995*; *Tyler and Pozzo-Miller, 2001*), exposure to proBDNF leads to smaller dendrites with fewer synapses (*Koshimizu et al., 2009*; *Yang et al., 2014*). Furthermore, the precise balance of BDNF and proBDNF is essential for the organization of synapses into clusters during development (*Kirchner and Gjorgjieva, 2021*; *Winnubst et al., 2015*; *Kleindienst et al., 2011*; *Niculescu et al., 2018*). Interestingly, synaptic activity triggers the cleaving of proBDNF into BDNF (*Je et al., 2012*), providing a mechanistic link between the molecular factors driving dendrite maturation and neural activity.

Activity-dependent factors are equally important for driving dendritic growth. As the sensory periphery is immature during early postnatal development, when many dendrites grow (*Leighton and Lohmann, 2016*), many developing circuits generate their own spontaneous activity. The rich spatiotemporal structure of spontaneous activity instructs the formation, removal, and change in the strength of synaptic inputs (*Sretavan et al., 1988*; *Sakai, 2020*) and triggers the stabilization or retraction of entire dendritic branches (*Riccomagno and Kolodkin, 2015*; *Lohmann et al., 2002*). Although blocking spontaneous activity does not result in a grossly different dendrite morphology, the density and specificity of synaptic connections are strongly perturbed (*Campbell et al., 1997*; *Ultanir et al., 2007*), highlighting the instructive effect of spontaneous activity on dendritic development (*Crair, 1999*).

An influential hypothesis tying the extrinsic signaling factors underlying dendritic development is the synaptotrophic hypothesis (*Vaughn, 1989*). According to this hypothesis, a growing dendrite preferentially extends to regions where it is likely to find synaptic partners. Once a dendrite finds such a partner, a synaptic contact forms, anchors the developing dendrite, and serves as an outpost for further dendritic growth. In contrast, loss of synaptic input to the dendrite can lead to retraction unless the remaining synapses stabilize the branch (*Lohmann et al., 2002*; *Niell et al., 2004*; *Haas et al., 2006*; *Cline and Haas, 2008*; *Riccomagno and Kolodkin, 2015*; *Cline, 2016*). However, elaborate dendrites with morphologically defined synapses can also emerge without any synaptic transmission (*Verhage et al., 2000*; *Cijsouw et al., 2014*; *Ferreira Castro et al., 2020*), suggesting that synaptic activity influences dendritic growth, but it is not the only driving force. Despite significant interest in the synaptotrophic hypothesis, we still lack a mechanistic understanding of how activity-dependent and -independent factors combine to shape development.

To investigate interactions between known signaling factors and synthesize information from different experimental results, computational models of dendrite development provide a fruitful direction to explore how different mechanisms can generate realistic dendritic morphologies (*Cuntz, 2016*; *Ferreira Castro et al., 2020*). Previous approaches include modeling dendritic development with random branching (*Kliemann, 1987*) or as a reaction-diffusion system (*Luczak, 2006*), implementing activity-independent growth cones that detect molecular gradients (*van Veen and van Pelt, 1992*; *Torben-Nielsen and De Schutter, 2014*), or building dendrites as the solution to an optimal wiring problem (*Cuntz et al., 2010*). Although these approaches are capable of generating dendritic structures that closely replicate the statistical properties of both developing and mature biological dendrites (*Koene et al., 2009*; *Cuntz, 2016*), they provide limited insight into the interaction between dendritic growth, synaptogenesis, and the local activity-dependent organization of synaptic inputs. Consequently, the association between morphological variability and electrophysiological (*Gouwens et al., 2020*; *Scala et al., 2021*) or functional (*Poirazi and Mel, 2001*; *Poirazi et al., 2003*; *Park*

*et al., 2019*; *Poirazi and Papoutsi, 2020*) characteristics of synaptic and dendritic attributes remains obscure.

Here, we propose a mechanistic computational model for cortical dendritic development for dendrite growth and synapse formation, stabilization, and elimination based on reciprocal interactions between activity-independent growth signals and spontaneous activity. Starting from neuronal somata distributed in a flat sheet of the cortex, spatially distributed potential synapses drive the growth of stellate-like dendrites through elongation and branching by activity-independent cues. Upon contact, synaptic connections form and stabilize or disappear according to a local activity-dependent learning rule inspired by neurotrophin interactions based on correlated patterns of spontaneous activity (*Kirchner and Gjorgjieva, 2021*). Consistent with the synaptotrophic hypothesis, the stability of a dendritic branch depends on the stability of its synaptic contacts, and the branch likely retracts after substantial synaptic pruning. The resulting dynamic system naturally leads to the emergence of three distinct phases of dendrite development: (1) an initial overshoot phase characterized by dendrite growth and synapse formation, (2) a pruning phase during which the learning rule prunes poorly synchronized synapses, and (3) a stabilization phase during which morphologically stable dendrites emerge from the balancing of growth and retraction. Varying model parameters in biologically realistic ranges produces changes in dendrite length and synapse density consistent with experiments. Our mechanistic model generates dendrites with approximately optimal wiring length, which is a widely used criterion for evaluating dendritic morphology (*Cuntz et al., 2010*; *Cuntz et al., 2012*; *Chklovskii et al., 2002*). At the same time, the model leads to the activity-dependent emergence of functional synaptic organization and input selectivity. Therefore, our mechanistic modeling framework for the growth and stabilization of dendritic morphologies and the simultaneous synaptic organization is ideally suited for making experimental predictions about the effect of perturbing specific model components on the resulting dendritic morphologies and synaptic placement.

## Results

We built a computational model of activity-dependent dendrite growth during development based on synapse formation, stabilization, and elimination. We focused on the basal stellate-like dendrites of cortical pyramidal neurons, which primarily extend laterally within a layer of the cortex (*Larkman and Mason, 1990*) and receive numerous feedforward and recurrent inputs (*Rossi et al., 2019*; *Iacaruso et al., 2017*). Stellate morphologies are found in many types of neurons, especially in the somatosensory cortex, including interneurons and layer 4 spiny stellate cells, which are the main recipients of thalamic input and play a key role in sensory processing (*Schubert et al., 2003*; *Marques-Smith et al., 2016*; *Scala et al., 2019*). To investigate the impact of synapse formation on dendrite development, we modeled several neuronal somata and potential synapses in a flat sheet of cortex (*Figure 1a*). Potential synapses represent locations in the cortex where an axon can form a synapse with a nearby dendrite (*Stepanyants and Chklovskii, 2005*). The model consists of two components: an activity-independent component that directly governs branch growth and retraction, and an activity-dependent component that governs synaptic organization and thus indirectly affects branch stability. Inspired by the synaptotrophic hypothesis (*Vaughn, 1989*), we mimicked the effect of activity-independent molecular signaling by letting each potential synapse release diffusive signaling molecules that attract the growing dendrite (*Figure 1b*, *Figure 1—figure supplements 1 and 2*). In addition, during development and before the onset of sensory experience, neural circuits generate patterned spontaneous activity (*Blankenship and Feller, 2010*; *Ackman and Crair, 2014*). Therefore, to model the structured spontaneous activity exhibited by different axons (*Scholl et al., 2017*; *Iacaruso et al., 2017*), we randomly divided the potential synapses into different activity groups that receive inputs correlated within a group but uncorrelated between the groups (see Materials and methods). Each group represents synapses from the same presynaptic neuron or neurons that experience correlated presynaptic activity.

Attracted by synapses which release growth factors and independent of neural activity, dendrites in our model extend from the soma toward the closest potential synapse, where they establish a synapse and temporarily stabilize (*Figure 1b*, *Figure 1—figure supplement 2*). We assumed that the dendrites could not overlap based on experimental data *Grueber and Sagasti, 2010*; therefore, the dendrites in the model retract, e.g., when further growth would require self-overlap. Once a synapse is formed, we modeled that its strength changes according to a local activity-dependent plasticity rule

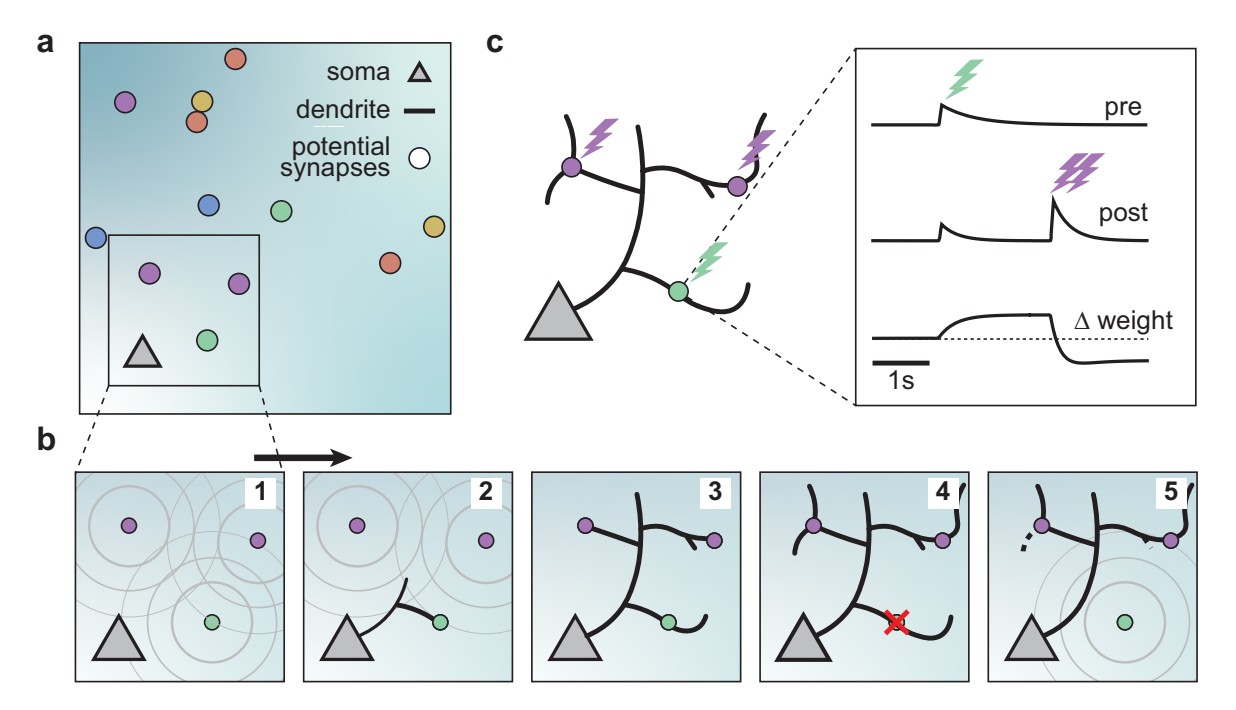

**Figure 1.** A model of dendritic growth for a cortical pyramidal neuron driven by activity-independent and -dependent mechanisms. (**a**) Schematic of the soma of a pyramidal neuron (orange triangle) with 12 randomly distributed potential synapses from presynaptic axons (circles) with correlated activity patterns indicated by color. (**b**) Schematic of activity-independent and -dependent mechanisms. Soma and synapses correspond to box in a. Signaling molecules diffusing from potential synapses (1) attract dendrite growth and promote synapse formation (2) independent of firing pattern (3). Over time, poorly synchronized synapses depress and are pruned from the dendrite (4), while well-synchronized synapses remain stable (5). After a branch retracts, the dendrite is less sensitive to the growth field at that location (5). (**c**) Change in weight of one synapse (green) following the stimulation of itself (green bolt) and of two nearby synapses (purple bolts). Left: Schematic of the developing dendrite from b with bolts indicating synaptic activation. Right: Presynaptic accumulator (top), postsynaptic accumulator (middle), and change in synaptic weight (bottom) as a function of time (see Materials and methods, *Kirchner and Gjorgjieva, 2021* for details of the plasticity rule). Dashed line (bottom) indicates zero change.

The online version of this article includes the following figure supplement(s) for figure 1:

**Figure supplement 1.** The growth field is similar to two-dimensional heat diffusion.

**Figure supplement 2.** Detailed illustration of the dendritic growth mechanism.

(*Kirchner and Gjorgjieva, 2021*; *Figure 1c*). The learning rule induces synaptic potentiation when presynaptic and local postsynaptic activity coincide. Conversely, it induces synaptic depression when local postsynaptic activity occurs in the dendrite independently of presynaptic stimulation, usually due to the activation of a neighboring synapse (see Materials and methods and the 'offset' constant below),

$$\Delta \text{weight} = \text{post} \times (\text{pre} - \text{offset}). \tag{1}$$

As shown previously, this rule generates synaptic distance-dependent competition, where nearby synapses affect each other more than distant synapses, and correlation-dependent cooperation, where neighboring synchronized synapses stabilize. In contrast, neighboring desynchronized synapses depress (*Kirchner and Gjorgjieva, 2021*). In our model, we assumed that when a synapse depresses below a threshold, it decouples from the dendrite, and the corresponding branch retracts successively to the nearest stable synapse, branch point, or the soma (*Figure 1b*, *Figure 1—figure supplement 2*). After removal, the vacated synapse turns into a potential synapse again, attracting other growing branches. Thus, a developing dendrite in our model acquires its arborization through the attraction to signaling molecules released by potential synapses and the repeated activity-dependent formation, stabilization, and removal of synapses.

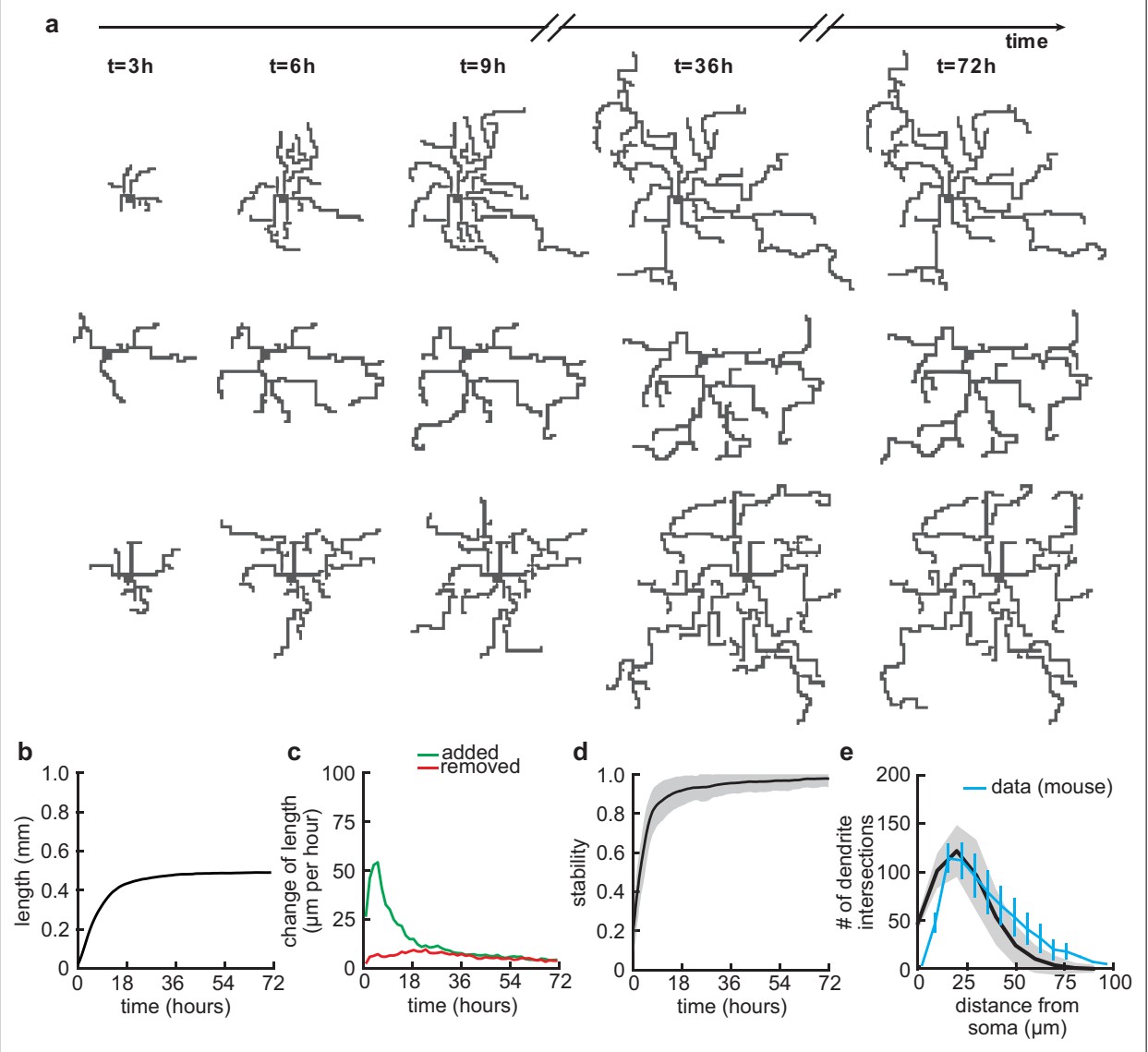

**Figure 2.** Balanced growth and retraction generate morphologically stable dendrites. (**a**) Three example dendrites at five time points from our simulations. For clarity of presentation, connected synapses are not displayed. (**b**) Total length of dendritic tree as a function of time. (**c**) Length of dendrite added (green) and removed (red) as a function of time. (**d**) Morphological stability (correlation between the dendrite shape at time $t$ and $t - 4.5$ hr) as a function of time. (**e**) Average number of dendrite intersections as a function of distance from the soma (the Sholl diagram). Data from basal dendrites in the developing mouse medial prefrontal cortex superimposed, normalized to the maximum (blue; *Kroon et al., 2019*). All lines represent averages across 32 simulations with nine dendrites each. Shaded area indicates two standard deviations.

The online version of this article includes the following video for figure 2:

**Figure 2—video 1.** Example of a simulation in which several dendrites develop in parallel.

https://elifesciences.org/articles/87527/figures#fig2video1

## Dendrite development through balancing growth and retraction

After specifying the rules governing the growth of individual dendritic branches, we investigated dendritic development on longer timescales. When growing dendrites according to our proposed growth rule based on the attraction of signaling molecules and spontaneous activity-dependent synaptic refinements (*Figure 1*), we found that dendrites form several stems, i.e., branches which start directly at the soma, and rapidly expand outward (*Figure 2a*). After an initial phase of rapid expansion, we observed that growth rapidly attenuates and dendritic length stabilizes (*Figure 2b*). This stability is achieved when the expansion and retraction of the dendrite are balanced (*Figure 2c*). To investigate

**Table 1.** Parameters of the minimal plasticity model (***Kirchner and Gjorgjieva, 2021***) and the synaptotrophic growth model.

| Parameter | Variable | Value |
| --- | --- | --- |
| Synaptic efficacy time constant | $\tau_W$ | 3000 time steps |
| Postsynaptic accumulator time constant | $\tau_u$ | 300 time steps |
| Presynaptic accumulator time constant | $\tau_v$ | 600 time steps |
| Constitutive percent of BDNF of total neurotrophins | $\eta$ | 46.5% |
| MMP9 efficiency constant | $\phi$ | 3/50 per time step |
| Heterosynaptic offset | $\rho$ | $\rho = \dfrac{2\eta - 1}{2(1 - \eta)}$ |
| Minimal model synaptic efficacy time constant | $\tau_w$ | $\tau_w = \tau_W \dfrac{1}{2(1 - \eta)}$ |
| Standard deviation of calcium spread | $\sigma_c$ | 200 μm |
| Turnover threshold below which a synapse is replaced | $W_{\text{thr}}$ | 0.02 |
| Firing rate of synapses | $r_{total}$ | 0.116 μm min$^{-1}$ |
| Scout intervals and speed | $t_{scout}, v_{scout}$ | 10 min, 0.18 μm min$^{-1}$ |

whether the stability in total length also corresponds to stability in dendritic morphology, we quantified morphological stability as the pixel-wise correlation of a dendrite with itself 4.5 hr earlier, which is several orders of magnitude larger than the speed at which dendrites grow and retract in our model (see *Table 1*). Despite the residual amount of expansion and retraction, we showed that morphological stability increases rapidly and the dendritic morphology is already stable after the initial expansion phase (*Figure 2d*). Interestingly, such rapid stabilization of morphology has also been observed in the mouse visual cortex (***Richards et al., 2020***) and the *Drosophila* larvae (***Ferreira Castro et al., 2020***). Next, we quantified the Sholl diagram, the number of dendritic branches at a given distance from the soma, commonly used as a measure of dendritic complexity (***Sholl and Uttley, 1953***; ***Binley et al., 2014***; ***Bird and Cuntz, 2019***). The Sholl diagram of stabilized dendrites generated by our model is unimodal and qualitatively matches the Sholl diagram of developing basal dendrites from the mouse medial prefrontal cortex (*Figure 2e*; data extracted from ***Kroon et al., 2019***, postnatal days 6–8), as well as the hippocampus (***Kleindienst et al., 2011***). In summary, by combining activity-independent and -dependent dendritic growth mechanisms, our model produces dendrites that rapidly expand and stabilize by balancing growth and retraction.

## Delayed activity-dependent plasticity produces a rapid increase in synapse density followed by pruning

Since our model couples dendritic growth with the formation and removal of synapses (*Figure 3a*), we next investigated how the number of connected synapses, which are necessary for the dendrite's stabilization, changes over time. As a result of the dendrite's rapid growth, we observed a rapid increase in the number of connected synapses (*Figure 3b and c*). In contrast to dendritic elongation, our findings indicate that the initial rapid increase in the number of connected synapses is subsequently followed by a transient phase of net synapse reduction before reaching a balance of synaptic additions and removals (*Figure 3c*). This removal of established synapses resembles the postnatal removal of synapses observed in the mouse neocortex (***Holtmaat et al., 2005***). To understand how the initial overshoot and subsequent removal of synapses emerge in our model, we computed the average synaptic weight of all synapses that eventually stabilize, or are pruned (*Figure 3d*). We found that the delayed onset of synapse removal (*Figure 3c*) is due to the slow timescale of synaptic weight

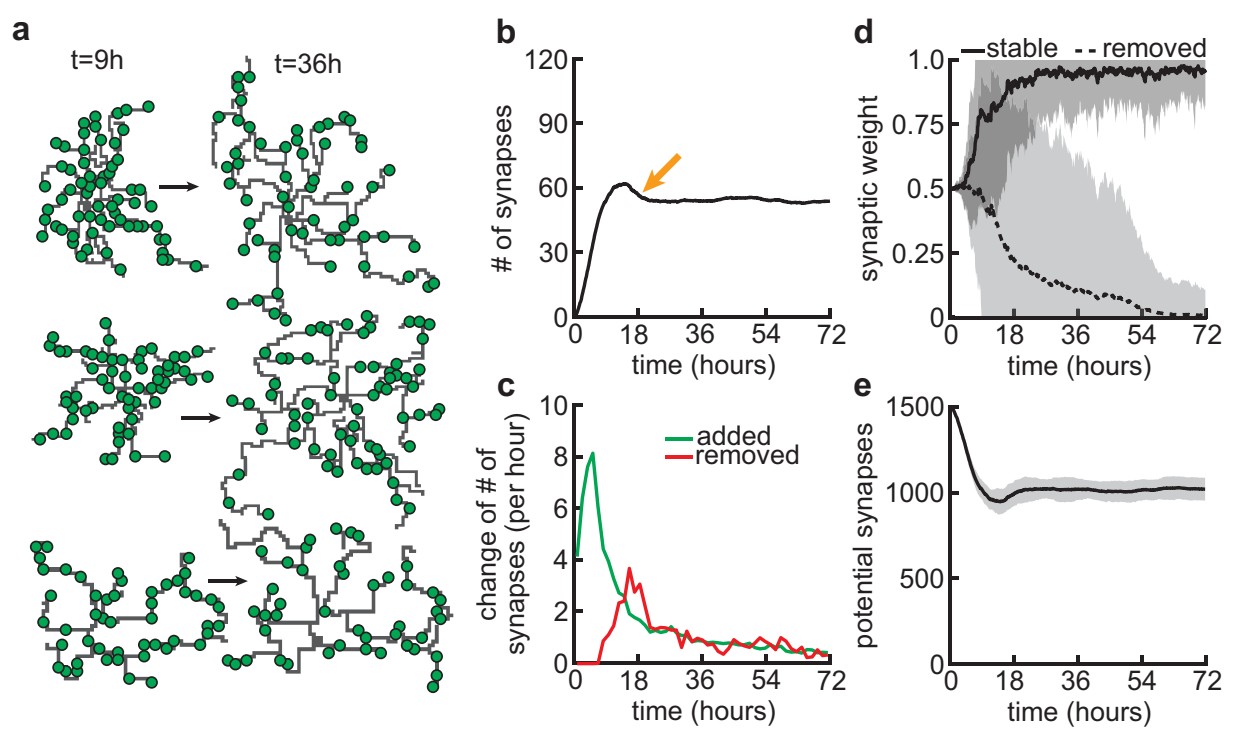

**Figure 3.** Synapse formation and removal predominate in distinct phases of dendrite development. (**a**) Three examples of dendrites at the beginning ($t = 9$ hr) and end ($t = 72$ hr) of the simulation. Green circles indicate formed synapses. (**b**) Total number of connected synapses as a function of time. Orange arrow highlights overshoot and subsequent pruning. (**c**) Added (green) and pruned synapses (red) as a function of time. (**d**) Average synaptic weights of synapses that ultimately stabilize (solid black; final weight more than $0.5$) or are removed (dashed black; final weight less than $0.5$) as a function of time. (**e**) Number of available potential synapses as a function of time. All lines represent averages across 32 simulations with nine dendrites each. Shaded area indicates two standard deviations.

The online version of this article includes the following figure supplement(s) for figure 3:

**Figure supplement 1.** Dendritic arbors stay stable over longer simulations.

**Figure supplement 2.** Somata density does not affect the timing of dendritic growth and pruning.

**Figure supplement 3.** Dynamic potential synapses with log-normally distributed lifetimes.

changes compared to the faster timescale of dendrite growth. Thus, the initial overshoot and subsequent removal of synapses observed in our model (*Figure 3b*) is due to the rapid formation relative to the delayed activity-dependent elimination of synapses. After the initial overshoot and pruning, dendritic branches in the model stay stable, with mainly small subbranches continuing to be refined (*Figure 3—figure supplement 1*). This stability in the model is achieved despite the number of potential synaptic partners remaining high (*Figure 3e*), indicating a balance between activity-independent and activity-dependent mechanisms. The dendritic growth and synaptic refinement dynamics are independent of the postsynaptic somata densities used in our simulations (*Figure 3—figure supplement 2*). Only the final arbor size and the number of connected synapses decrease with an increase in the density of the somata, while the timing of synaptic growth, overshoot, and pruning remains the same (*Figure 3—figure supplement 2*).

The development of axons is concurrent with dendritic growth and highly dynamic (*Matsumoto et al., 2024*). To address the impact of simultaneously growing axons, we implemented a simple form of axonal dynamics by allowing changes in the lifetime and location of potential synapses, originating from the axons of presynaptic partners (*Figure 3—figure supplement 3*). When potential synapses can move rapidly (median lifetime of 1.8 hr), the model dynamics are perturbed quite substantially, making it difficult for the dendrites to stabilize completely (*Figure 3—figure supplement 3c*). However, slowly moving potential synapses (median lifetime of 18 hr) still yield comparable results (*Figure 3—figure supplement 3*). The distance of movement significantly influenced results only when potential synaptic lifetimes were short. For extended lifetimes, the moving distance had a minor

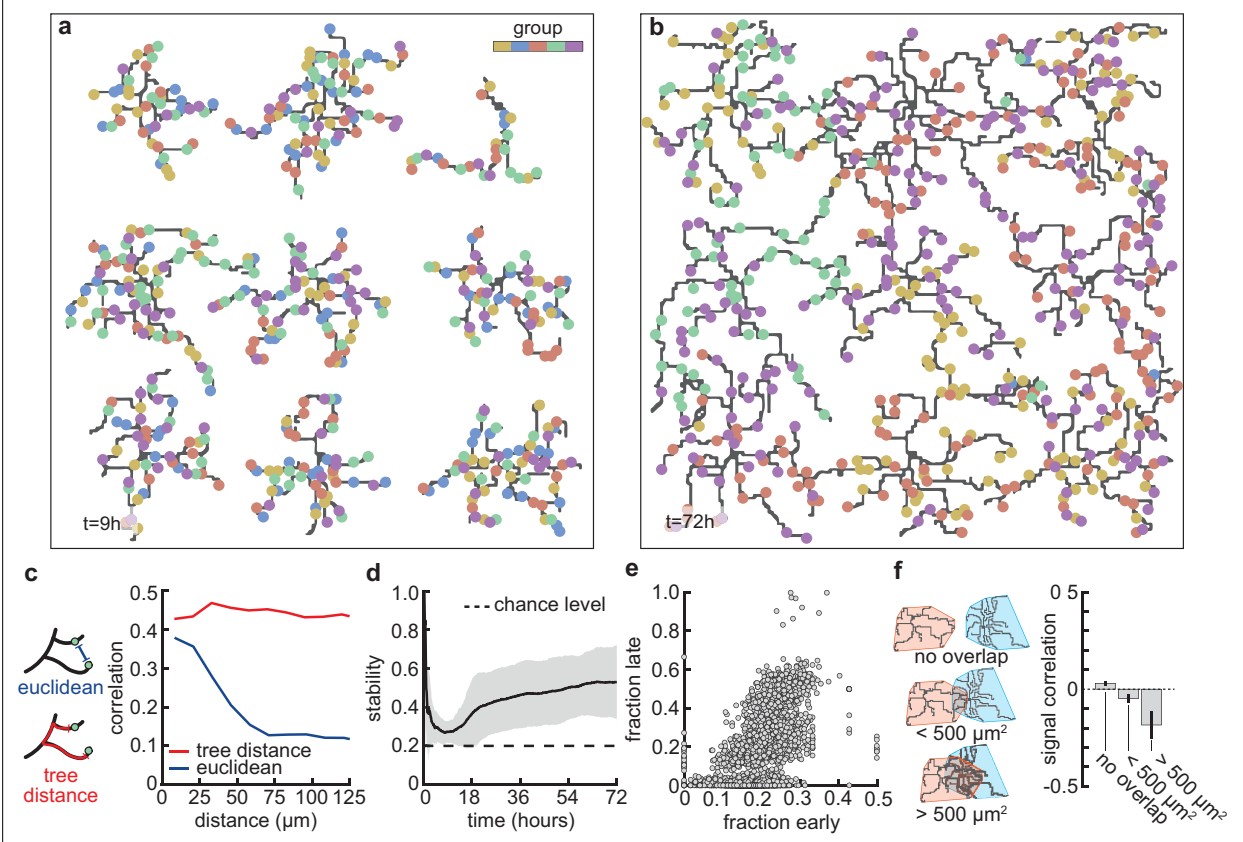

**Figure 4.** Stable morphology is obtained through selective removal of synapses and dendritic input selectivity. (**a, b**) Dendritic trees before (a, 9 hr) and after (b, 72 hr) removal of synapses (*Figure 3*). Connected synapses colored corresponding to activity group, which represents activity correlations (*Figure 1b*). (**c**) Left: Schematic illustrating the difference between Euclidean and tree distance. Note that we compute the Euclidean distance between synapses from different trees. Right: Correlation between pairs of synapses as a function of the Euclidean distance (blue) and tree distance (red). (**d**) Input selectivity of dendrites (defined as the fraction of the activity group with the highest representation) as a function of time. Dashed line indicates chance level. All lines represent averages across 32 simulations with nine dendrites each. Shaded area indicates two standard deviations. (**e**) Fraction of connected synapses per activity group early ($t = 9$ hr) and late ($t = 72$ hr) in the simulation. Each dot represents one of the five activity groups on one of the nine dendrites from the 32 simulations, resulting in $5 \times 9 \times 32 = 1440$ data points. (**f**) Left: Schematic of different levels of overlap (rows) between the convex hulls of two dendrites, referring to the smallest convex sets that contain the dendrite. Right: Signal correlation (correlation between fractions of synapses from the same activity groups) for different levels of dendritic overlap. Error bars indicate the standard error of the mean, computed from 1152 pairs of dendrites from 32 simulations.

impact on the dynamics, predominantly affecting the time required for dendrites to stabilize. This was the result of synapses being able to transfer from one dendrite to another, potentially forming new long-lasting connections even at advanced stages of synaptic refinement. In sum, our results show that potential axonal dynamics only affect the stability of our model when these dynamics are much faster than dendritic growth.

## Activity-dependent competition between synapses produces input selectivity and synaptic organization

Next, we asked whether the stabilization of dendrites might be supported by the emergence of organization of connected synapses. First, we compared the synapses connected to dendrites at the apex of the overshoot phase (peak in *Figure 3b*) with those in the stabilization phase (*Figure 4a and b*). While dendrites at the apex do not prefer synapses from any particular input group, in the stabilization phase, they acquire a preference for synapses from only two or three of the activity groups (*Figure 1b*). These dynamics resemble activity-dependent synaptic competition in the developing visual cortex, where asynchronously activated synapses tend to depress (*Winnubst et al., 2015*). In particular, the remaining synchronized synapses in our model experience correlation-dependent

cooperation (*Kirchner and Gjorgjieva, 2021*), and therefore are able to stabilize the dendrite and prevent the total retraction of all branches.

This selective potentiation of synapses according to input correlation also leads to dendritic selectivity for inputs. In particular, synapses on the same dendrite are likely to be of the same activity group (*Figure 4c*). This selectivity is acquired throughout the simulation, where selectivity starts high (a nascent dendrite is trivially selective to its first synapse; $t = 0 - 1$ hr), drops almost to chance level (indiscriminate addition of synapses; $t = 9$ hr), and finally approaches a value of $\frac{1}{2}$ (two activity groups per dendrite remain after the pruning phase; $t = 72$ hr) (*Figure 4d*). To determine which activity group eventually stabilizes, we computed selectivity for each group early ($t = 9$ hr) and late ($t = 72$ hr). We found that early high (low) selectivity for an activity group translates into even higher (lower) selectivity in the stable state (*Figure 4e*), predicting an outsized impact of early synaptic contacts on continued dendritic function. Finally, we observed that when dendritic trees overlap strongly, they tend to be selective to different activity groups (*Figure 4f*) due to competition for limited potential synapses of any given group. Interestingly, also in the mouse visual cortex, neighboring neurons often exhibit different selectivity (*Ohki et al., 2005*), potentially reflecting the lasting impact of early competition between different inputs.

In summary, the emergence of dendrite selectivity for synapses of specific activity groups coincides with and supports the stabilization of dendritic morphologies.

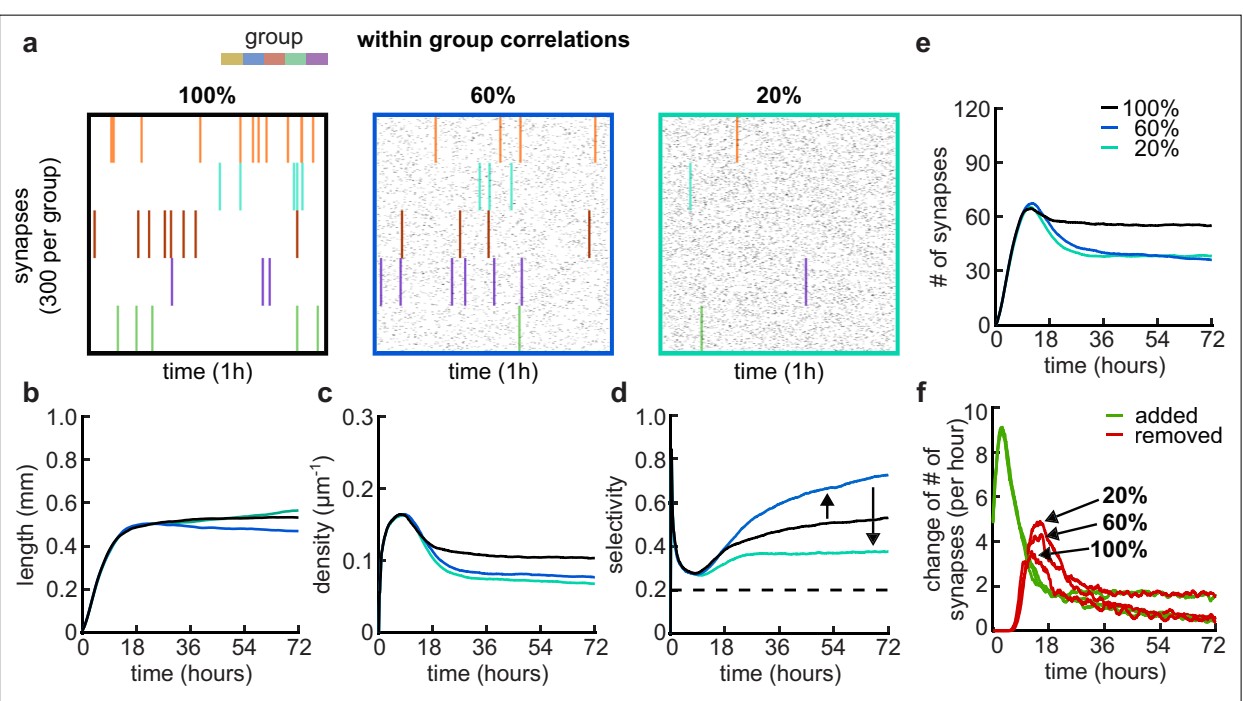

**Figure 5.** Effects of within-group correlations on group selectivity and synaptic pruning dynamics. (**a**) Correlated spike train examples of all the activity groups with different within-group correlation statistics (from left to right, 100%, 80%, and 20% within-group correlations). Spike trains are sorted by synapses that correspond to each of the activity groups. Correlated spikes that are shared across all the synapses of one group are highlighted by the group's color. (**b**) Total dendritic length as a function of time. (**c**) Total density of synapses on the dendrite. (**d**) Input selectivity of dendrites (defined as the fraction of the activity group with the highest representation) as a function of time. Dashed line indicates chance level. (**e**) Total number of connected synapses as a function of time. (**f**) Added (green) and pruned synapses (red) as a function of time. Arrows indicate the peaks of the simulations with different correlations. All lines represent averages across 32 simulations with nine dendrites each.

The online version of this article includes the following figure supplement(s) for figure 5:

**Figure supplement 1.** Examples of coexisting activity groups and synaptic pruning with low within-group correlations.

**Figure supplement 2.** Effects of across-group correlations and combinations of within- and across-group correlations on group selectivity and synaptic pruning dynamics.

## Group activity correlations shape synaptic overshoot and selectivity competition across synaptic groups

Since correlations between synapses emerge from correlated patterns of spontaneous activity abundant during postnatal development (*Ackman, 2012*; *Siegel et al., 2012*), we explored a wide range of within-group correlations in our model (*Figure 5a*). Although a change in correlations within the group has only a minor effect on the resulting dendritic lengths (*Figure 5b*) and overall dynamics, it can change the density of connected synapses and thus also affect the number of connected synapses to which each dendrite converges throughout the simulations (*Figure 5c and e*). This is due to the change in specific selectivity of each dendrite which is a result of the change in within-group correlations (*Figure 5d*). While it is easier for perfectly correlated activity groups to coexist within one dendrite (*Figure 5—figure supplement 1a*, 100%), decreasing within-group correlations increases the competition between groups, producing dendrites that are selective for one specific activity group (60%, *Figure 5d*, *Figure 5—figure supplement 1a*). This selectivity for a particular activity group is maximized at intermediate (approximately 60%) within-group correlations, while the contribution of the second most abundant group generally remains just above random chance levels (*Figure 5—figure supplement 1a*). Further reducing within-group correlations causes dendrites to lose their selectivity for specific activity groups due to the increased noise in the activity patterns (20%, *Figure 5a*). Overall, reducing within-group correlations increases synapse pruning (*Figure 5f*, bottom), also found experimentally (*Matsumoto et al., 2024*), as dendrites require an extended period to fine-tune connections aligned with their selectivity biases. This phenomenon accounts for the observed reduction in both the density and number of synapses connected to each dendrite.

In addition to the within-group correlations, developmental spontaneous activity patterns can also change correlations between groups as, for example, retinal waves propagated in different domains (*Feller et al., 1997*; *Figure 5—figure supplement 2*). An increase in between-group correlations in our model intuitively decreases competition between the groups since fully correlated global events synchronize the activity of all groups (*Figure 5—figure supplement 2*). The reduction in competition reduces pruning in the model, which can be recovered by combining cross-group correlations with decreased within-group correlations (*Figure 5—figure supplement 2*). Our simulations show that altering the correlations in presynaptic activity increases competition (by lowering the within-group correlations) or decreases competition (by raising the across-group correlations). Hence, in our model, competition between activity groups due to non-trivially structured correlations is necessary to generate realistic dynamics between activity-independent growth and activity-dependent refinement or pruning.

In sum, our simulations demonstrate that our model can operate under various correlations in the spike trains. We find that the level of competition between synaptic groups is crucial for the activity-dependent mechanisms to either potentiate or depress synapses and is fully consistent with recent experimental evidence showing that the correlation between spontaneous activity in retinal ganglion cells axons and retinal waves in the superior colliculus governs branch addition vs. elimination (*Matsumoto et al., 2024*).

## Balance of mature and immature BDNF controls the arborization of dendrites

After establishing that our model can capture important aspects of the dynamics of dendritic development through the combination of activity-independent and activity-dependent mechanisms, including local plasticity, we asked how changing properties of the plasticity rule might affect dendritic growth and synaptic organization. Developmentally, the interaction between two neurotrophic factors, BDNF and proBDNF (*Figure 6a*), has been found to play a key role in the organization of synaptic inputs into clusters (*Niculescu et al., 2018*). Therefore, through the previously established link between this neurotrophin interaction and synaptic plasticity (*Kirchner and Gjorgjieva, 2021*), we investigated the influence of changing the underlying molecular interactions on dendritic morphology.

As we have previously shown, the 'offset' term in our plasticity rule (*Equation 1*) represents the neurotrophin balance (computed as BDNF/(BDNF+proBDNF)) released upon stimulation of a synapse (*Kirchner and Gjorgjieva, 2021*). Consequently, we found that an overabundance of BDNF (proBDNF) leads to potentiation (depression) of the synapse (*Figure 6b*), consistent with experimental data (*Lu et al., 2005*). Furthermore, our plasticity rule acts locally on the dendrite, so that the strength of

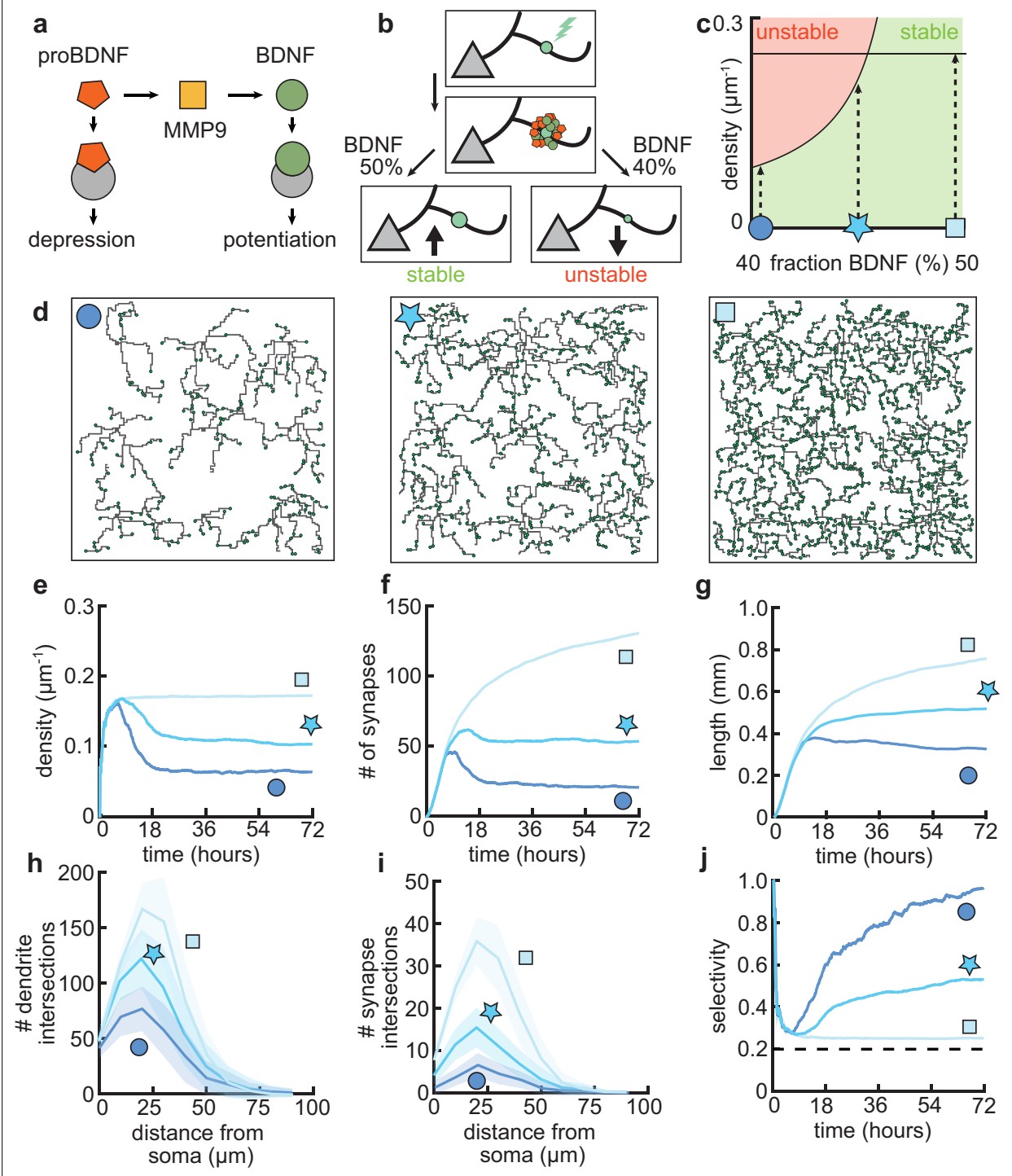

**Figure 6.** Dendritic arborization is controlled by the ratio of neurotrophic factors. (**a**) Interactions between molecular factors underlie a local activity-dependent plasticity rule for synaptic change (*Equation 1*, *Kirchner and Gjorgjieva, 2021*). Neurotrophins (brain-derived neurotrophic factor [BDNF] and proBDNF) bind to different neurotrophin receptors, and a cleaving protease (MMP9) converts proBDNF to BDNF in an activity-dependent manner. (**b**) Schematic illustrating the impact of different concentrations of BDNF on synaptic change. Upon stimulation of a synapse (top), proBDNF and BDNF is released into extracellular space (middle), where proBDNF can be cleaved into BDNF by MMP9. Depending on the neurotrophin ratio, computed as BDNF/(BDNF+proBDNF), the synapse is stabilized (left) or depressed and hence eventually removed (right). (**c**) Maximally possible stable density of synapses as a function of the initial concentration of BDNF. Stable (no pruning; green) and unstable (pruning occurs; red) areas are indicated. (**d**) Three examples of dendrites with superimposed synapses (green) with high initial BDNF concentration (49%), the baseline concentration (46.5%, same as *Figure 1–Figure 3*), and low initial BDNF (43%). Symbols correspond to locations marked in panel c. (**e–g**) Averages for density of synapses on the

*Figure 6 continued on next page*

*Figure 6 continued*

dendrite (**e**), number of connected synapses (**f**) and total length of dendrite (**g**) as a function of time for dendrites from the three conditions shown in d. (**h–i**) Average number of dendrite intersections (**h**) and synapses (**i**) as a function of distance from the soma for dendrites from the three conditions shown in d. (**j**) Global selectivity as a function of time for dendrites from the three conditions shown in d. Dashed line indicates chance level. All lines represent averages across 32 simulations with nine dendrites each. Shaded area indicates two standard deviations.

individual synapses is affected by interactions with other nearby synapses. Concretely, a lower (higher) density of nearby synapses tends to lead to potentiation (depression) of the synapse (*Kirchner and Gjorgjieva, 2021*).

To better understand the interactions between the balance of neurotrophins and the density of synapses, we analytically derived the maximum density of synapses that can stabilize given a balance of neurotrophins (*Figure 6c*, see Materials and methods). We found that an overabundance of BDNF (proBDNF) leads to a higher (lower) maximal density of synapses (*Figure 6c*). Indeed, when we simulated dendritic development with varying neurotrophin ratios, we found that the density of synapses per dendrite increases with increasing neurotrophin ratio (*Figure 6d and e*). Consistent with biological evidence (*McAllister et al., 1995*; *Tyler and Pozzo-Miller, 2001*), in our model, developing dendrites treated with BDNF tend to grow larger and have a higher density of synapses (*Figure 6e and g*). In contrast, over-expression of proBDNF leads to smaller dendrites with fewer synapses (*Koshimizu et al., 2009*; *Yang et al., 2014*; *Figure 6f and g*). Perturbing the balance between neurotrophins scales the Sholl diagram of dendrite intersections and synapses, but does not qualitatively affect the shape of the curve (*Figure 6h and i*).

In our model, these changes in length and density are explained by a change in the selectivity of the synapses (*Figure 6j*). Concretely, an increase in BDNF erases all synaptic competition, reducing the selectivity to chance level, while an increase in proBDNF greatly amplifies synaptic competition and thus selectivity. These differences in competition determine the number of pruned synapses and thus the length at which the dendrite stabilizes. Thus, our model predicts that biologically relevant differences in dendritic morphology may arise from different neurotrophin ratios due to the maximal density of synapses that stabilizes the dendrite.

## Different impacts of activity-dependent and -independent factors on dendritic development

Our mechanistic model enabled us to dissect the different roles of activity-dependent and -independent mechanisms on dendritic morphology. To this end, we varied only activity-dependent factors or only activity-independent factors across a set of simulations (*Figure 7a*). We introduced variability in the activity-dependent aspects of the model through the firing patterns of potential synapses, and in the activity-independent aspects of the model via fluctuations in both the extrinsic growth signals and the intrinsic mechanisms underlying dendrite growth (see Materials and methods, *Figure 7b*).

Consistent with experiments (*Scala et al., 2021*), dendrites produced by our model exhibit substantial variability in morphology (*Figure 7a*), length (*Figure 7c*), and number of synapses (*Figure 7d*). Comparing dendrites that experienced either identical activity-dependent or identical activity-independent factors allowed us to compute the percentage of change in morphology attributable to each factor as a function of developmental time (*Figure 7e and f*). We found that while activity-independent factors tend to lead to large differences in morphology early on, activity-dependent factors affect dendrite morphology with a substantial delay. These differences can be explained by the delay in synaptic pruning relative to initial synaptic formation (*Figure 3d*).

Despite substantial variability, there are predictive factors for the final length of the dendrite. In particular, we found a positive relationship between the number of major branches, i.e., branches starting from the soma and the final length (*Figure 7g*). Interestingly, this is consistent with dendrites reconstructed from multiple regions of the mouse cortex (*Figure 7—figure supplement 1*). Furthermore, our model predicts that dendrites that have a high (low) total length early on will, on average, retain a (high) low total length throughout development (*Figure 7e*).

Thus, our model suggests that while activity-independent factors affect dendritic morphology early on during development, activity-dependent factors dominate later. Properties such as the number of major branches or the length of dendrites during early development might be predictive of the dendrite's morphology throughout the animal's lifetime.

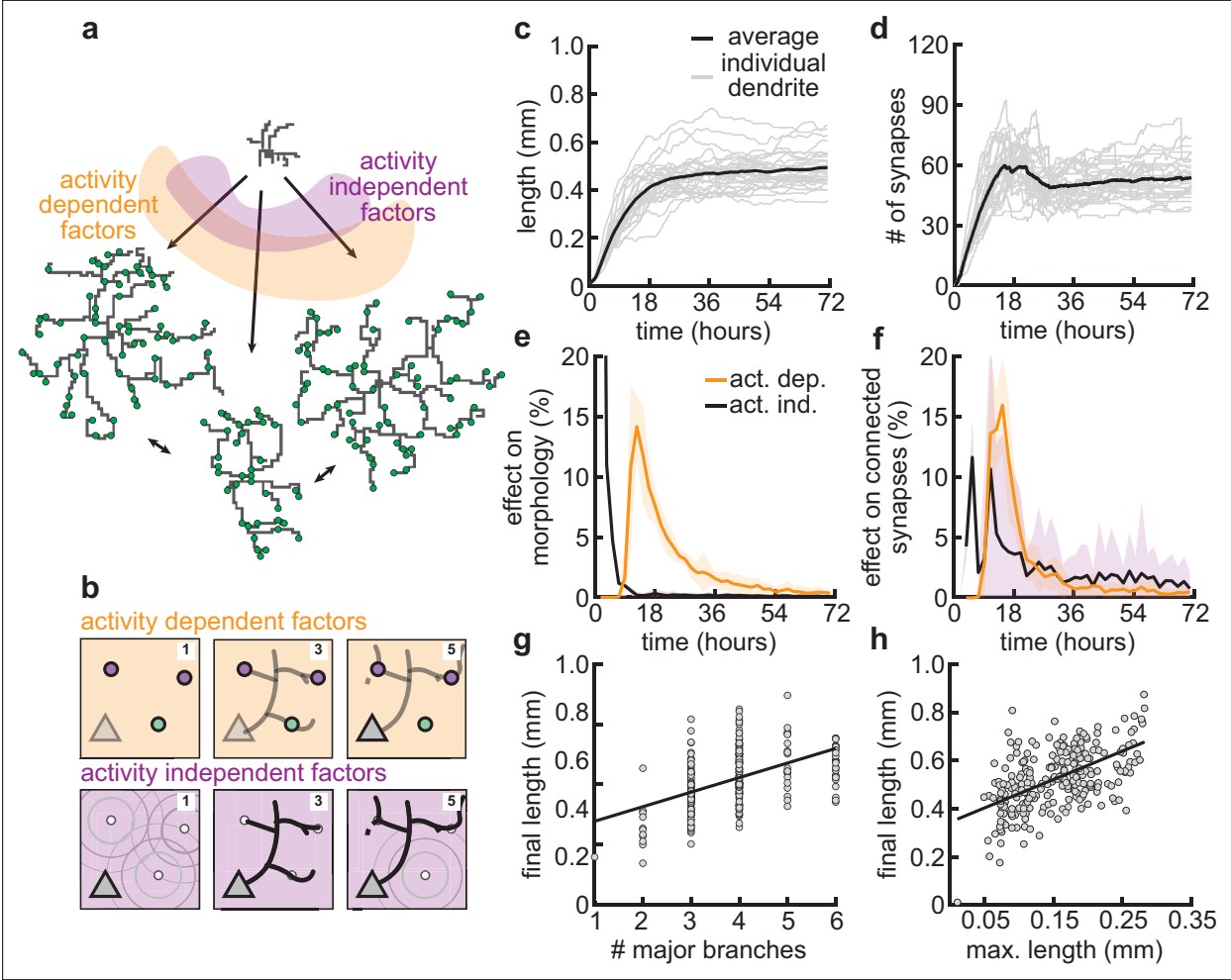

**Figure 7.** Morphological variability emerges from the interaction of activity-dependent and -independent factors. (**a**) Example of three dendrites with identical initial conditions but different random seeds. The colors illustrate that initial growth is governed by activity-independent factors, while later growth is governed by activity-dependent factors. (**b**) Schematic illustrating how variability is introduced into model: activity-dependent via the patterns of spontaneous activity (orange), and activity-independent via fluctuations in both the extrinsic growth stimulating field (purple 1) and the intrinsic mechanisms underlying dendrite growth (purple 2; see Materials and methods). (**c, d**) Total length (**c**) and number of synapses (**d**) as a function of time for dendrites with identical initial conditions but different random seeds. Each gray line corresponds to one dendrite from 1 of 32 simulations. Bold line represents average. (**e, f**) Percentage in change of morphological similarity (**e**) and similarity of connected synapses (**f**) as a function of time for simulations where activity-dependent (orange) or -independent (purple) factors vary. Lines represent averages across 32 simulations with nine dendrites each. Shaded area indicates two standard deviations. (**g, h**) Final length as a function of number of major branches (**g**) and maximal length in the first 18 hr of the simulation (**h**). Lines indicate linear regression.

The online version of this article includes the following figure supplement(s) for figure 7:

**Figure supplement 1.** Total tree length increases with the number of stems.

## Coupled dendrite growth and synapse formation leads to approximately optimal minimal wiring

Given the spatial constraints within cortical architecture and the significant metabolic costs associated with sustaining complex dendritic morphologies, it is advantageous for neurons to establish synaptic connections using the shortest possible dendritic length (*Cuntz et al., 2012*; *Figure 8a*, top vs. middle). Yet another constraint that neurons must adhere to as brain size increases is the need to minimize conduction delays (*Figure 8a*, bottom). A balance between these two conservation laws constrains dendrite morphology (*Cuntz et al., 2010*; *Cuntz et al., 2012*; *Ferreira Castro et al., 2020*). In particular, in our model, dendrites are assumed to grow toward the nearest potential

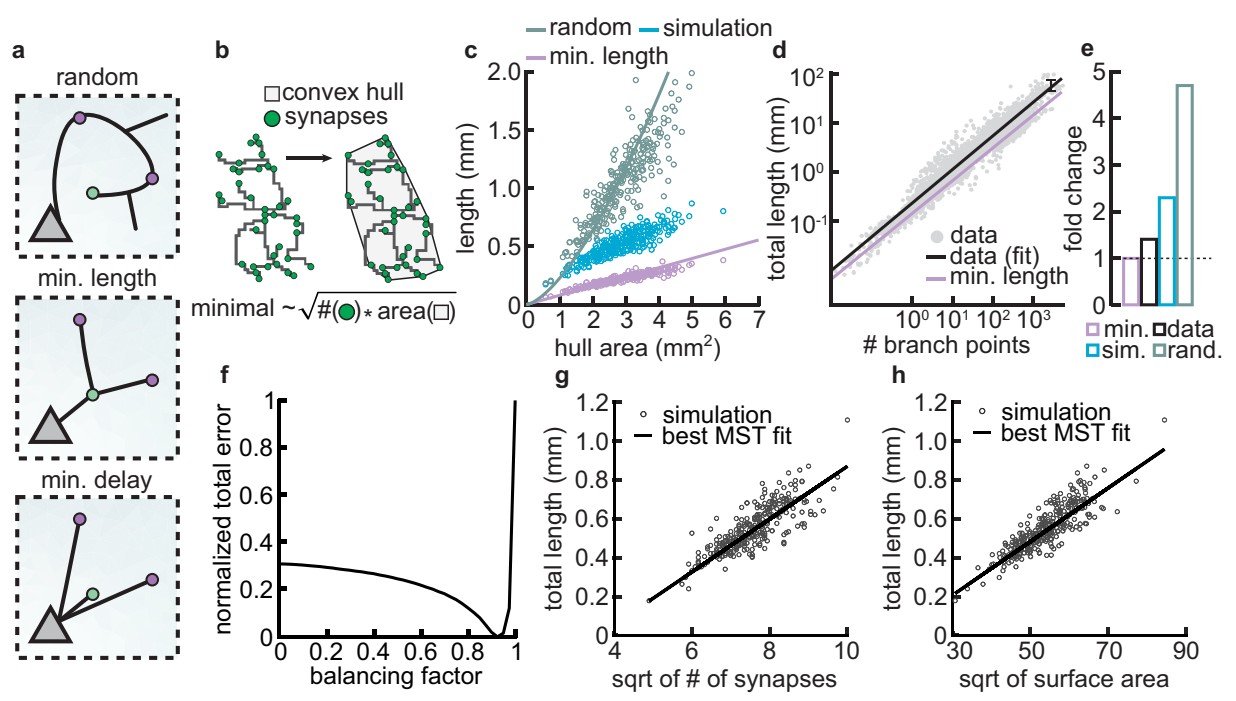

**Figure 8.** Synthetic dendritic morphologies: analysis of optimal wiring principles. (**a**) Schematic of synaptic wiring principles: random (top), minimal length (middle), and minimal delay (bottom) for a set of synapses. (**b**) The convex hull of all synapses connected to a dendrite with the proportionality of minimal wiring length (bottom). (**c**) Total tree length as a function of convex hull area in the minimal wiring length, simulated and random scenario. Each dot corresponds to one of 288 dendrites from 32 simulations. Lines correspond to analytic predictions for the average density across all simulations. (**d**) Total tree length against the number of branch points in log scale, both for data and theoretical minimum. Data extracted from *Cuntz et al., 2012*. (**e**) Total tree length in the data (black, average of *n*=13,112), our simulations (blue, average of 288 dendrites from 32 simulations), and the random baseline (green, analytically computed) relative to the theoretical minimum (pink, analytically computed). (**f**) Model parameter *bf* vs. error (comparing simulated total length, number of terminals, and surface area, with corresponding synthetic trees; straight black line; optimal *bf* = 0.9250). (**g–h**) Scaling behavior of the square root of number of synapses against total length (g, $R^2 = 0.65$) and square root of surface area against total length (h, $R^2 = 0.73$) showing the relationships expected from the optimal wire equations. The black line shows the best minimum spanning tree (MST) *bf* fit scaling behavior (see Materials and methods). In both panels, each dot represents one synthetic tree (*n*=288). Lines were fit using linear regression.

synapse. Thus, we investigated how the final dendritic length produced by our models compares to the principles of optimal wiring.

As an initial analysis, our aim was to assess the upper and lower bounds to which our models could be constrained. As a lower limit for wiring minimization, we used known scaling relationships where the length (*L*) of a minimally wired dendrite in a plane scales with the square root of the number of synapses (*N*) times the area over which the synapses are distributed (*A*): $L = \sqrt{NA/\pi}$ (*Cuntz et al., 2012*). In contrast, as an upper bound, the length of a dendrite when synapses are connected randomly scales with the number of connected synapses times the average distance of two random points on a circle (*Uspensky, 1937*), $L = N\frac{128}{45\pi}\sqrt{A/(2\pi)}$. This approximation assumes a circular spanning domain, which simplifies the calculation of mean pairwise distances. However, for non-circular domains, such as highly eccentric or irregular shapes, the mean pairwise distance can be larger, leading to an underestimation of the upper bound. This differs from the optimal wiring solution by a square root in the number of synapses. Using the convex hull that circumscribes the stabilized synapses as the area over which the synapses are distributed (*Figure 8b*), we compared the actual length of the dendrite with minimal and the random wiring length (*Figure 8c*). Our analysis reveals that the dendritic lengths generated by our simulations show values that are shorter compared to those predicted by random wiring models, yet they are longer than those predicted by the theoretical minimum length.

Next, we wanted to know if the deviation from minimal wiring might quantitatively match the one observed in real dendrites. To investigate this question, we reanalyzed a published dataset (*Cuntz et al., 2012*) containing the total lengths and the number of branch points of 13,112 dendrites pooled across 74 sources. When computing the fold change between the real and the theoretical expectations

derived from the scaling law in the dataset, we confirmed that real dendrites tend to be consistently larger than the theoretical minimum (*Figure 8d*). Interestingly, the fold change between the length of real dendrites and the theoretical minimum is similar to the fold change of our simulated dendrites and the theoretical minimum (*Figure 8e*). This deviation is expected given that real dendrites need to balance their growth processes between minimizing wire while reducing conduction delays. The interplay between these two factors emerges from the need to reduce conduction delays, which requires a direct path length from a given synapse to the soma, consequently increasing the total length of the dendritic cable (*Cuntz et al., 2010*; *Cuntz et al., 2012*; *Ferreira Castro et al., 2020*).

To investigate this further, we compared the scaling relations of the final morphologies of our models with other synthetic dendritic morphologies generated using a previously described minimum spanning tree (MST) based model. The MST model balances the minimization of total dendritic length and the minimization of conduction delays between synapses and the soma. This balance results in deviations from the theoretical minimum length because direct paths to synapses often require longer dendrites (*Cuntz et al., 2008*; *Cuntz et al., 2010*). The balance in the model is modulated by a balancing factor (*bf*). If *bf* is zero, dendritic trees minimize the cable only, and if *bf* is one, equal weight is given to minimizing conduction delays and total dendritic length. Although *bf* is not inherently bounded by 1 in the cited studies, these cases serve to illustrate how different weightings influence the resulting morphologies. It is important to note that the MST model does not simulate the developmental process of dendritic growth; it is a phenomenological model designed to generate static morphologies that resemble real cells.

To facilitate the comparison of total lengths between our simulated and MST morphologies, we generated MST models under the same initial conditions (synaptic spatial distribution) as our models and simulated them to match several morphometrics (total length, number of terminals, and surface area) of our grown morphologies. This allowed us to create a corresponding MST for each of our synthetic trees. Consequently, we could evaluate whether the branching structures of our models were accurately predicted by MSTs based on optimal wiring constraints. We found that the best match occurred with a trade-off parameter $bf = 0.9250$ (*Figure 8f*). Using the morphologies generated by the MST model with the specified trade-off parameter (*bf*), we showed that the square root of the synapse count and the total length ($L$) in both of our model-generated trees and the MSTs exhibit a linear scaling relationship (*Figure 8g*; $R^2 = 0.65$). The same linear relationship can be observed for the square root of the surface area and the total length $L$ of our model trees and the MSTs (*Figure 8h*; $R^2 = 0.73$). Overall, these results indicate that our model-generated trees are well fitted by the MST model and follow wire optimization constraints.

We acknowledge that the value of the balancing factor *bf* in our model is higher than the range of balancing factors that is typically observed in the biological dendritic counterparts, which are specifically planar (two-dimensional) and generally ranges between 0.2 and 0.4 (*Cuntz et al., 2012*; *Ferreira Castro et al., 2020*; *Baltruschat et al., 2020*). However, it is still remarkable that our model, which does not explicitly address these two conservation laws, achieves approximately optimal wiring. Why do we observe such a high *bf* value? We reason that two factors may contribute to this. First, in our models, local branches grow directly to the nearest potential synapse, potentially taking longer routes instead of optimally branching to minimize wiring length (*Wen and Chklovskii, 2008*). Second, the growth process in our models does not explicitly address the tortuosity of the branches, which can increase the total length of the branches used to connect synapses. In the future, it will be interesting to add constraints that take these factors into account. Taken together, combining activity-independent and -dependent dendrite growth produces morphologies that approximate optimal wiring.

## Discussion

Dendritic growth and the formation, stabilization, and removal of synapses during early development depend on various factors, including extrinsic factors such as growth cues, intrinsic molecular signaling, and correlated patterns of spontaneous activity. However, the nature of these interactions and the implications for dendritic function throughout life remain largely unexplored. In this study, we proposed a mechanistic model for the growth and retraction of dendritic branches, as well as the formation and removal of synapses on these dendrites during development, based on the interaction of activity-independent cues from potential synaptic partners and local activity-dependent synaptic

plasticity. Our model can simultaneously capture two main aspects of dendritic growth: producing dendritic morphologies and driving synaptic organization.

## Assumptions and predictions of the model

Some of the most prominent models of dendritic growth have focused on activity-independent rules based on geometric or biophysical constraints (*Cuntz et al., 2010*; *Cuntz et al., 2012*). Despite their immense success in generating realistic dendritic morphologies, they leave open the question of the underlying biological mechanisms. Other studies have implemented global activity-dependent rules that require feedback signals from the soma or the axon (*Van Ooyen et al., 1995*). Our model proposes a simple and biologically plausible mechanism for the simultaneous dendritic growth and synaptic organization based on activity-independent cues and local activity-dependent learning rules, which cluster synaptic inputs according to functional preferences. Numerous experimental studies have demonstrated the importance of such local plasticity for the emergence of local synaptic organization in the form of clusters, as well as dendritic function (*Hering and Sheng, 2001*; *Lohmann et al., 2002*; *Chen et al., 2013*; *Niculescu et al., 2018*).

Our model makes some simplifying assumptions at the expense of mechanistic insights. For instance, we model the generation of only stellate-like morphologies without the apical trunk. Many types of neurons are characterized by stellate morphologies, especially in the somatosensory cortex (*Schubert et al., 2003*; *Marques-Smith et al., 2016*; *Scala et al., 2019*). Nonetheless, it would be interesting to investigate if our model's mechanisms can be minimally modified to apply to the generation of apical dendrites. Moreover, we generate our model dendrites in a two-dimensional, flat sheet of cortex. We anticipate that the models can be straightforwardly extended to three dimensions, but with additional computational cost. Although our assumptions may be too simplified to generate perfectly biologically realistic morphologies, the simple rules in our model capture basic morphological features, such as the number of branches, the total length, and the Sholl analysis, with those of biological neurons reported in the literature.

A key advantage of our mechanistic model is the ability to predict the impact of early perturbations on mature dendritic morphology, as the model allows us to independently investigate activity-independent and -dependent influences on dendrite growth and synaptic organization. For example, three distinct phases of synapse development, overshoot, pruning, stabilization, and stable dendritic trees, emerge naturally from the interactions between activity-independent signaling and activity-dependent synaptic plasticity, without additional assumptions and are robust to parameter variations. The stabilization of dendritic morphologies in our model is enabled by the emergence of input selectivity, which implies local organization of synapses responsive to a particular correlated input pattern on the dendrite. Hence, our model explains how dendritic morphology can adapt to changes in activity-dependent plasticity or input statistics during development, as observed experimentally (*Cline and Haas, 2008*; *McAllister et al., 1995*; *Tyler and Pozzo-Miller, 2001*; *Matsumoto et al., 2024*). Further, we provide a mechanistic explanation for the emergence of approximately optimal minimal wiring in cortical dendrites. Thus, our model provides a new perspective on the interaction of activity-independent and -dependent factors influencing dendrite growth and suggests that the formation and activity-dependent stabilization vs. removal of synapses might exert powerful control over the growth process.

The correlated activity experienced by our modeled synapses (and resulting synaptic organization) does not necessarily correspond to visual orientation, or any stimulus feature, for that matter, but is rather a property of spontaneous activity. Nonetheless, there is some variability in what the experimental data show. Many have shown that synapses on dendrites are organized into functional synaptic clusters: across brain regions, developmental ages, and diverse species from rodent to primate (*Kleindienst et al., 2011*; *Winnubst et al., 2015*; *Iacaruso et al., 2017*; *Scholl et al., 2017*; *Niculescu et al., 2018*; *Takahashi et al., 2012*; *Gökçe et al., 2016*; *Wilson et al., 2016*; *Kerlin et al., 2019*; *Ju et al., 2020*; *Hedrick et al., 2022*; *Hedrick et al., 2024*). Other studies have reported lack of fine-scale synaptic organization (*Chen et al., 2013*; *Varga et al., 2011*; *Chen et al., 2011*; *Jia et al., 2010*; *Jia et al., 2014*). Interestingly, some of these discrepancies might be explained by different species showing clustering with respect to different stimulus features (orientation or receptive field overlap) (*Scholl et al., 2017*; *Wilson et al., 2016*; *Iacaruso et al., 2017*). Our prior work proposed a theoretical framework to reconcile these data: combining activity-dependent plasticity

as we used in the current work, and a receptive field model for the different species (*Kirchner and Gjorgjieva, 2021*).

## Comparison with the synaptotrophic hypothesis

The synaptotrophic hypothesis, originally proposed three decades ago (*Vaughn, 1989*), has provided a useful framework for interpreting the effect of neural activity and synapse formation on dendrite development. Our proposed model is inspired by the synaptotrophic hypothesis, but differs from it in a few key aspects. (1) The synaptotrophic hypothesis postulates that synaptic activity is necessary for dendrite development (*Cline and Haas, 2008*). In contrast, our model contains an activity-independent component that allows dendrites to grow even in the absence of synaptic activity. Our model is thus consistent with the finding that even in the absence of neurotransmitter secretion connected neuronal circuits with morphologically defined synapses can still be formed (*Verhage et al., 2000*; *Ferreira Castro et al., 2023*) and with computational (non-mechanistic) models that produce dendrites with many relevant morphological properties without relying on activity (*Cuntz, 2016*). (2) The synaptotrophic hypothesis does not specify the exact molecular factors underlying the information exchange pre- and postsynaptically. Based on recent experiments that identify central molecular candidates (*Winnubst et al., 2015*; *Kleindienst et al., 2011*; *Niculescu et al., 2018*; *Lu et al., 2005*), our model proposes a concrete mechanistic implementation based on neurotrophic factors (*Kirchner and Gjorgjieva, 2021*). (3) The synaptotrophic hypothesis postulates that whether a potential synaptic contact is appropriate can be rapidly evaluated pre- and postsynaptically. Inspired by experiments (*Lohmann et al., 2002*; *Niell et al., 2004*), the fate of a synapse in our model is determined only within tens of minutes or hours after it is formed. This is due to the slow timescale of synaptic plasticity.

Although we did not explicitly model postsynaptic firing, our previous work with static dendrites has shown that it can play an important role in establishing a global organization of synapses on the entire dendritic tree of the neuron (*Kirchner and Gjorgjieva, 2021*). For example, we showed that it could lead to the emergence of retinotopic maps on the dendritic tree which have been found experimentally (*Iacaruso et al., 2017*). Since we use the same activity-dependent plasticity model in this paper, we expect that the somatic firing will have the same effect on establishing synaptic distributions on the entire dendritic tree.

## Relationship between dendritic form and function

In contrast to previous studies which focused on how dendritic morphology affects function, e.g., through nonlinear signal transformation (*Poirazi and Papoutsi, 2020*) or dynamic routing of signals (*Payeur et al., 2019*), we propose that dendrite form and function reciprocally shape each other during development. While the morphology of the dendrite constrains the pool of available potential synapses, synaptic activity determines the dendritic branch's stability (*Figure 1*). As a result, the dendritic tree self-organizes in an appropriate shape to support a limited number of functionally related synapses. These initial synaptic contacts might then serve as a scaffold around which additional functionally related synapses cluster to form the building blocks to support the powerful computations of mature dendrites (*Kirchner and Gjorgjieva, 2022*).

## Dynamics of dendritic development

Here, we focus on the early developmental period of highly dynamic dendritic growth and retraction. However, dendritic morphology remains remarkably stable in later development and throughout adulthood (*Richards et al., 2020*; *Ferreira Castro et al., 2020*; *Koleske, 2013*). This stability is achieved despite substantial increases in the overall size of the animal (*Richards et al., 2020*; *Ferreira Castro et al., 2020*) and ongoing functional and structural plasticity of synapses (*Kleindienst et al., 2011*; *Winnubst et al., 2015*; *Kirchner and Gjorgjieva, 2021*). Although it is still unclear how exactly synaptic organization is established during early development and how synapses are affected by the overall increase in dendrite size, somatic voltage responses to synaptic activity are largely independent of dendrite size (*Cuntz et al., 2021*). The stability of dendrites has been shown to play a crucial role in allowing proper function of the adult brain and is affected in many psychiatric disorders and neurodegenerative diseases. In particular, the release of BDNF, which is related to synaptic activity, affects structural consolidation of dendrites and therefore long-term stability (*Koleske, 2013*). Our mechanistic model allows us to perturb the balance of neurotrophic factors and investigate the effects

on dendritic development. For instance, our model predicts detrimental effects on dendrite stability as a result of extreme or non-existent input selectivity, providing insights into functional consequences of disrupted dendrite growth in neurodevelopmental disorders (*Johnston et al., 2016*).

### Interneurons and inhibitory synapses

In addition to excitatory neurons and synapses that are the focus of this study, inhibitory interneurons and inhibitory synapses also play an important role in brain development (*Naskar et al., 2019*). Interneurons fall into genetically distinct subtypes, which tend to target different portions of pyramidal neurons (*Rudy et al., 2011*; *Kepecs and Fishell, 2014*). In particular, somatostatin-expressing interneurons preferentially target the dendrites of pyramidal neurons, while parvalbumin-expressing interneurons preferentially target the soma. Furthermore, the dendrites of inhibitory neurons have a complex morphology that likely allows them to perform intricate transformations of incoming signals (*Tzilivaki et al., 2019*; *Tzilivaki et al., 2022*). Investigating whether and how inhibitory interneurons and synapses might interact with excitatory ones during dendritic development is an exciting direction for future research.

In summary, by proposing a mechanistic model of dendritic development that combines activity-independent and activity-dependent components, our study explains several experimental findings and makes predictions about the factors underlying variable dendritic morphologies and synaptic organization. Interestingly, the stable morphologies it generates are approximately optimal in terms of wiring length and experimental data. Finally, our model provides the basis for future exploration of different learning rules and cell types which could differ across brain regions, species, and healthy vs. disease states.

## Materials and methods

### Activity-independent synaptic signals

In the synaptotrophic hypothesis, dendrite growth is directed toward potential synaptic partners. In our model, we captured this aspect by introducing a *growth field* of activity-independent synaptic signals, $T(\mathbf{p})$, over all positions $\mathbf{p}$ in our sheet of cortex. This field contains point sources at the positions of potential synapses, $\mathbf{p}_i$, and evolves over time according according to a diffusion equation,

$$T(\mathbf{p})^{t+1} = T(\mathbf{p})^t * D + \mu \sum_i \mathbf{p}_i + \sigma \mathbf{N}. \tag{2}$$

The growth field at time point $t + 1$ is therefore given by the sum of the growth field at time $t$ convolved with a diffusion filter $D$, a constant input of size $\mu$ from all potential synapses, which are currently not connected to a dendrite, as well as independent Gaussian noise, $\mathbf{N}$, with standard deviation $\sigma$. We chose a two-dimensional Gaussian for the diffusion filter $D$, making the field $T(\mathbf{p})$ mathematically equivalent to a heat diffusion in two dimensions (*Figure 1—figure supplement 1*).

### Asynchronous dendrite growth and retraction

Dendrite development critically depends on resources from the soma (*Ye et al., 2007*). Consequently, we modeled the growth of dendrites to depend on *scouting agents* that spread outward from the soma at regular time intervals, $t_{scout}$, and that traverse the dendritic tree at speed $v_{scout}$ (*Figure 1— figure supplement 2*). These scouting agents resemble actin-blobs that control dendrite growth (*Nithianandam and Chien, 2018*). When a scouting agent encounters a branch point, there is a 0.5 chance for it to continue in any direction. This means it can go in one direction, but it can also duplicate or disappear completely. We further modeled these scouting agents to detect the growth field's value – a scouting agent stops at a position on the dendrite where this field is locally maximal and marks this location for growth. The dendrite will then expand at the marked positions in the direction of the gradient of the growth field, and the scouting agent moves to this new position. If the dendrite grows to the location of a potential synapse, this synapse is then realized, and its initial weight is set to $w_{init} = \frac{1}{2}$. Two branches of the same neuron may never become adjacent; however, branches from other neurons may be crossed freely. If a scouting agent reaches the end of a branch without finding a local maximum of the growth field along its path, the scouting agent initiates the retraction

of this branch. Depending on proximity, a branch then retracts up to the nearest stable synapse, the next branch point, or the soma. Because our simulations are a spatially discrete approximation of a biological flat sheet of cortex, we had to ensure that growth behaves appropriately in cases where the discretization scheme becomes relevant (*Figure 1—figure supplement 2*).

## Minimal plasticity model

When a synapse $k$ forms on the dendrite, its weight $w_k$ evolves according to a previously proposed minimal plasticity model for interactions between synapses on developing dendrites (*Kirchner and Gjorgjieva, 2021*). This model can be linked to a full neurotrophin model that interprets the parameters in terms of the neurotrophic factors BDNF, proBDNF, and the protease MMP9. In this model, the $k$-th synapse is stimulated within input event trains $x_k$

$$x_k(t) = \int_0^\infty \sum_f \delta(s - s_k^f)(H(t - s) - H(t - x_{\mathrm{dur}} - s))\mathrm{d}s \tag{3}$$

with events at times $t_k^f$ and where the Heaviside step function $H(t)$ is 0 when $t$ is less than 0 and 1 when $t$ is greater or equal than 0, so that events have duration $x_{\mathrm{dur}}$ (50 time steps). The minimal plasticity model consists of a synapse-specific presynaptic accumulator $v_k$,

$$\tau_v \frac{\mathrm{d}v_k}{\mathrm{d}t} = -v_k(t) + \phi x_k(t), \tag{4}$$

and a postsynaptic accumulator $u_k$ that averages over nearby synapses in a weighted and distance-dependent manner,

$$\tau_u \frac{\mathrm{d}u_k}{\mathrm{d}t} = -u_k(t) + \sum_{l=1}^N s_{kl} w_l(t) x_l(t). \tag{5}$$

The multiplicative factor $\phi$ is an MMP9 efficiency constant that determines how efficiently MMP9 converts proBDNF into BDNF per unit of time and the proximity variables $s_{kl}$ between synapses $k$ and $l$ on the same dendrite are computed as $s_{kl} = e^{-\frac{d_{kl}^2}{2\sigma_s^2}}$, where $\sigma_s$ determines the spatial postsynaptic calcium spread constant. The equation governing the weight development of $w_k$ (*Equation 6*) is a Hebbian equation that directly combines the pre- and postsynaptic accumulator with an additional offset constant $\rho$,

$$\tau_w \dot{w}_k = u_k(t)(v_k(t) + \rho), \tag{6}$$

with $\rho = \frac{2\eta - 1}{2(1 - \eta)}$ and $\tau_w = \tau_W \frac{1}{2(1 - \eta)}$. Here, $\eta$ is the constitutive ratio of BDNF to proBDNF and $\tau_W = 3000$ ms. This model is minimal in the sense that it cannot be reduced further without losing either the dependence on correlation through the link to the BTDP rule or the dependence on distance.

To model structural plasticity, we implemented a structural plasticity rule inspired by *Holtmaat and Svoboda, 2009*, where each synapse whose efficacy falls below a fixed threshold $W_{\mathrm{thr}}$ is pruned from the dendrite.

## Simulations and parameters

For all simulations in this study, when not stated otherwise, we distributed nine postsynaptic somata at regular distances in a grid formation on a two-dimensional $185 \times 185$ pixel area, representing a cortical sheet (where 1 pixel = 1 μm, *Figure 4*). This yields a density of around 300 neurons per mm² (translating to around 5000 per mm³, where for 25 neurons in *Figure 3—figure supplement 2* this would be around 750 neurons per mm² or 20,000 per mm³). The explored densities are a bit lower than compared to neuron densities observed in adulthood (*Keller et al., 2018*). In the same grid, we randomly distributed 1500 potential synapses, yielding an initial density of 0.044 potential synapses per μm² (*Figure 3e*). At the end of the simulation time, around 1000 potential synapses remain, showing that the density of potential synapses is sufficient and does not significantly affect the final density of connected synapses. Thus, the rapid slowing down of growth in our model is not due to a depletion of potential synaptic partners. The resulting density of stably connected synapses is approximately

0.015 synapses per µm² (around 60 synapses stabilized per dendritic tree, *Figure 3b*). This density compares well to experimental findings, where, especially during early development, synaptic densities are described to be within a range similar to the one observed in our model (*Leighton et al., 2024*; *Ultanir et al., 2007*; *Glynn et al., 2011*; *Yang et al., 2014*; *Koshimizu et al., 2009*; *Tyler and Pozzo-Miller, 2001*). The potential synapses were divided into five groups of equal size (300 synapses per group), with each group receiving Poisson input with rate $r_{in}$. Therefore, all synapses in the same group are perfectly correlated, while synapses in different groups are uncorrelated.

Parameters, as the somata density, activity group correlations, or dynamic potential synapses, were only changed for simulations in which we explicitly explored their effects on the models behavior (see *Figure 3—figure supplement 2*, *Figure 3—figure supplement 3*, *Figure 5*, *Figure 5—figure supplement 2*, and sections below).

### Within-group and across-group activity correlations

For the decreased within-group correlations, we generated parent spike trains for each individual group with the firing rate $r_{in} = r_{total} * P_{in}$ (e.g. $P_{in} = 100\%; 60\%; 20\%$, *Figure 5*). All the synapses of the same group share the same parent spike train and the remaining spikes for each synapse are uniquely generated with the firing rate $r_{rest} = r_{total} * (1 - P_{in})$ (e.g. $(1 - P_{in}) = 0\%; 40\%; 80\%$), resulting in the desired firing rate $r_{total}$ (see *Table 1*). For the increase in across-group correlations, we generated one master spike train with the firing rate $r_{cross} = r_{total} * P_{cross}$ for all the synapses of all groups (e.g. $P_{cross} = 5\%; 10\%; 20\%$, *Figure 5—figure supplement 2*). This master spike train is shared across all groups and then filled up according to the within-group correlation (if not specified differently $P_{in} = 1 - P_{cross}$ to maintain the rate $r_{total}$). In all the cases, also in those where across-group correlations are combined with within-group correlations, the remaining spikes for each synapse are generated with a firing rate $r_{rest} = r_{total} * (1 - P_{in} - P_{cross})$, to obtain an overall desired firing rate of $r_{total}$.

### Dynamic potential synapses

To implement dynamic potential synapses we introduced lifetimes for each synapse, randomly sampled from a log-normal distribution with median 1.8 hr (for when they move frequently), 4.5 hr or 18 hr (for when they move rarely) and variance equal to 1 (*Figure 3—figure supplement 3b*). The lifetime of a synapse decreases only when the synapse is not connected to any of the dendrites (i.e. while it is a potential synapse). When the lifetime of a synapse expires, the synapse moves to a new location with a new lifetime sampled from the same log-normal distribution. This enables synapses to move multiple times throughout a simulation. The exact locations and distances to which each synapse can move are determined by a binary matrix (dimensions: *pixeldistance* × *pixeldistance*) representing a ring (annulus) with the inner radius $\frac{d}{4}$ and outer radius $\frac{d}{2}$, where the synapse location is at the center of the matrix. All the locations of the matrix within the ring boundaries (between inner radius and outer radius) are potential locations to which the synapse can move. The synapse then moves randomly to one of the possible locations where no other synapse or dendrite is located. For the movement distances, we chose the ring dimensions $3 \times 3$, $25 \times 25$, and $101 \times 101$, yielding the moving distances (radii) of 1 pixel per movement, 12 pixels per movement, and 50 pixels per movement ($r = (d - 1)/2$). These pixel distances represent small movements, as much as a dendrite can grow in one step (1 µm), and larger movements which are far enough so that the synapse will not attract the same branches again (12 µm) or far enough so that it might attract a completely different dendrite (50 µm, *Figure 3—figure supplement 3a*). Since the synapses only move inside the simulated area but never disappear, the total number of synapses remains constant througout the simulaiton.

### Comparison with wiring optimization MST models

To evaluate the wire minimization properties of our model morphologies (*n*=288), we examined whether the number of connected synapses (*N*), total length (*L*), and surface area of the spanning field (*S*) conformed to the scaling law $L \approx \pi^{-1/2} \cdot S^{1/2} \cdot N^{1/2}$ (*Cuntz et al., 2012*). Furthermore, to validate that our model dendritic morphologies scale according to optimal wiring principles, we created simplified models of dendritic trees using the MST algorithm with a balancing factor (*bf*). This balancing factor adjusts between minimizing the total dendritic length and minimizing the tree distance between synapses and the soma (Cost = $L + bf \cdot PL$) (MST_tree; best $bf = 0.925$) (*Cuntz et al., 2010*; TREES Toolbox http://www.treestoolbox.org). Initially, we generated MSTs to connect the same distributed

synapses as our models. We performed MST simulations that vary the balancing factor between $bf = 0$ and $bf = 1$ in steps of 0.025 while calculating the morphometric agreement by computing the error (Euclidean distance) between the morphologies of our models and those generated by the MST models. The morphometrics used were total length, number of terminals, and surface area occupied by the synthetic morphologies.

## Acknowledgements

This work was supported by the Max Planck Society and the European Research Council (ERC) under the European Union's Horizon 2020 research and innovation program (Grant agreement No. 804824 to JG). This work was also partially supported by a Daimler-Benz Foundation grant (AFC). We thank members of the 'Computation in Neural Circuits' group for discussions and feedback.

## Additional information

### Competing interests

Julijana Gjorgjieva: Reviewing editor, eLife. The other authors declare that no competing interests exist.

### Funding

| Funder | Grant reference number | Author |
|---|---|---|
| Max Planck Society | | Jan H Kirchner<br>Lucas Euler<br>Julijana Gjorgjieva |
| European Research Council | Grant agreement No. 804824 | Jan H Kirchner<br>Julijana Gjorgjieva |
| Daimler Benz Foundation | | Ingo Fritz<br>André Ferreira Castro |

The funders had no role in study design, data collection and interpretation, or the decision to submit the work for publication. Open access funding provided by Max Planck Society.

### Author contributions

Jan H Kirchner, Conceptualization, Software, Formal analysis, Investigation, Visualization, Methodology, Writing – original draft, Writing – review and editing; Lucas Euler, Software, Formal analysis, Investigation, Visualization, Methodology, Writing – original draft, Writing – review and editing; Ingo Fritz, Software, Investigation, Visualization, Methodology, Writing – original draft, Writing – review and editing; André Ferreira Castro, Formal analysis, Methodology, Writing – review and editing; Julijana Gjorgjieva, Conceptualization, Supervision, Funding acquisition, Investigation, Methodology, Writing – original draft, Project administration, Writing – review and editing

### Author ORCIDs

Lucas Euler http://orcid.org/0009-0004-2360-8104
Ingo Fritz http://orcid.org/0009-0005-3470-7105
Julijana Gjorgjieva https://orcid.org/0000-0001-7118-4079

Reviewer #2 (Public review): https://doi.org/10.7554/eLife.87527.3.sa1
Reviewer #3 (Public review): https://doi.org/10.7554/eLife.87527.3.sa2
Author response https://doi.org/10.7554/eLife.87527.3.sa3

# Additional files

## Supplementary files
MDAR checklist

## Data availability
The current manuscript is a computational study, so no data has been generated for this manuscript. All code can be accessed at: https://github.com/comp-neural-circuits/dendritic-growth (copy archived at *comp-neural-circuits, 2025*). The reanalysed dataset for Figure 8 panel d and e was taken from *Cuntz et al., 2012*.

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
