## [Editor Report · eLife Assessment]

This **important** work investigates how two distinct processes, morphological changes and synaptic plasticity, contribute to the final shape of neuronal dendrites and the spatial structure of their synaptic inputs. The modelling is **convincing** and could be broadly applied to other similar questions. The work will be of interest to neuroscientists studying dendritic development and connectivity at a single-cell level.

---

## [Referee Report · Reviewer #2 (Public review)]

This work combines a model of two-dimensional dendritic growth with attraction and stabilisation by synaptic activity. The authors find that constraining growth models with competition for synaptic inputs produces artificial dendrites that match some key features of real neurons both over development and in terms of final structure. In particular, incorporating distance-dependent competition between synapses of the same dendrite naturally produces distinct phases of dendritic growth (overshoot, pruning, and stabilisation) that are observed biologically and leads to local synaptic organisation with functional relevance. The approach is elegant and well-explained but makes some significant modelling assumptions that might impact the biological relevance of the results.

The main strength of the work is the general concept of combining morphological models of growth with synaptic plasticity and stabilisation. This is an interesting way to bridge two distinct areas of neuroscience in a manner that leads to findings that could be significant for both. The modelling of both dendritic growth and distance-dependent synaptic competition is carefully done, constrained by reasonable biological mechanisms, and well-described in the text. The paper also links its findings, for example in terms of phases of dendritic growth or final morphological structure, to known data well.

The authors have managed to address my previous comments on the paper well by considering axonal dynamics, spatial correlations, and the effects of changing ratios of BDNF-proBDNF. The modelling has now been validated over a wider range of confounding factors and looks to be a solid basis for future work in this direction.

---

## [Referee Report · Reviewer #3 (Public review)]

The authors propose a mechanistic model of how the interplay between activity-independent growth and an activity-dependent synaptic strengthening/weakening model influences the dendrite shape, complexity, and distribution of synapses. The authors focus on a model for stellate cells with multiple dendrites emerging from a soma. The activity-independent component is provided by a random pool of presynaptic sites representing potential synapses and releasing a diffusible signal promoting dendritic growth. Then, a spontaneous activity pattern with some correlation structure is imposed at those presynaptic sites. The strength of these synapses follows a learning rule previously proposed by the lab: synapses strengthen when there is correlated firing across multiple sites, and synapses weaken if there is uncorrelated firing with the relative strength of these processes controlled by available levels of BDNF/proBDNF. Once a synapse is weakened below a threshold, the dendrite branch at that site retracts and loses its sensitivity to the growth signal.

This revised version of the manuscripts contains clarifications and additional experiments that better reflect the robustness of the model. I continue to maintain my favorable review. I am glad the research persevered the long delays with changing trainees.

---

## [Author Response]

The following is the authors’ response to the original reviews.

**Public Reviews:**

**Reviewer #1 (Public Review):**
The authors introduce a computational model that simulates the dendrites of developing neurons in a 2D plane, subject to constraints inspired by known biological mechanisms such as diffusing trophic factors, trafficked resources, and an activity-dependent pruning rule. The resulting arbors are analyzed in terms of their structure, dynamics, and responses to certain manipulations. The authors conclude that (1) their model recapitulates a stereotyped timecourse of neuronal development: outgrowth, overshoot, and pruning (2) Neurons achieve near-optimal wiring lengths, and Such models can be useful to test proposed biological mechanisms- for example, to ask whether a given set of growth rules can explain a given observed phenomenon - as developmental neuroscientists are working to understand the factors that give rise to the intricate structures and functions of the many cell types of our nervous system.Overall, my reaction to this work is that this is just one instantiation of many models that the author could have built, given their stated goals. Would other models behave similarly? This question is not well explored, and as a result, claims about interpreting these models and using them to make experimental predictions should be taken warily. I give more detailed and specific comments below.

We thank the reviewer for the summary of the work. But the criticism “that this is one instantiation of many models [we] could have built” is unfair as it can apply to any model. We chose one of the most minimalistic models which implements known biological mechanisms including activity-independent and -dependent phases of dendritic growth, and constrained parameters based on experimental data. We compare the proposed model to other alternatives in the Discussion section. In the revised manuscript, we additionally investigate the sensitivity of model output to variations of specific parameters, as explained below.

Point 1.1. Line 109. After reading the rest of the manuscript, I worry about the conclusion voiced here, which implies that the model will extrapolate well to manipulations of all the model components. How were the values of model parameters selected? The text implies that these were selected to be biologically plausible, but many seem far off. The density of potential synapses, for example, seems very low in the simulations compared to the density of axons/boutons in the cortex; what constitutes a potential synapse? The perfect correlations between synapses in the activity groups is flawed, even for synapses belonging to the same presynaptic cell. The density of postsynaptic cells is also orders of magnitude of, etc. Ideally, every claim made about the model's output should be supported by a parameter sensitivity study. The authors performed few explorations of parameter sensitivity and many of the choices made seem ad hoc.

We have performed detailed sensitivity analysis on the model parameters mentioned by the reviewer, including (I) the density of postsynaptic cells (somatas), (II) the density of potential synapses, and (III) the level of correlations between synapses.

(I) While the density of postsynaptic cells in our baseline model seems a bit low, at least when compared to densities observed in adulthood (Keller et al., 2018), we explored how altering this value affects the model dynamics. We found that the postsynaptic cell density does not affect the timing of dendritic outgrowth, overshoot and synaptic pruning. It only changes the final size of the dendritic arbor and the resulting number of connected synapses. This analysis is now included in Supplementary Figure 3-2.

(II) The density of potential synapses and the density of connected synapses that we used in the manuscript are already in the range of densities that can be found in the literature (Leighton et al., 2024; Ultanir et al., 2007; Glynn et al., 2011; Yang et al., 2014), some of which we already cited in the original submission.

A potential concern might be that the rapid slowing down of growth in the model could be due to a depletion of potential synapses. To illustrate that this is not the case, we showed that the number of available potential synapses over the time course of the simulations remains high (Figure 3, new panel e). Therefore, the initial density of potential synapses is sufficient and does not affect the final density of connected synapses.

To further illustrate the robustness of our model dynamics to longer simulation times, we added a new supplementary figure (Supplementary Figure 3-1).

These new figure additions (Figure 3e, Supplementary Figure 3-1, and Supplementary Figure 3-2) and their implications for the model dynamics are discussed in the Results section of the revised paper:

p.9 line 198, “After the initial overshoot and pruning, dendritic branches in the model stay stable, with mainly small subbranches continuing to be refined (Figure 3-Figure Supplement 1). This stability in the model is achieved despite the number of potential synaptic partners remaining high (Figure 3e), indicating a balance between activity-independent and activitydependent mechanisms. The dendritic growth and synaptic refinement dynamics are independent of the postsynaptic somata densities used in our simulations (Figure 3-Figure Supplement 2). Only the final arbor size and the number of connected synapses decrease with an increase in the density of the somata, while the timing of synaptic growth, overshoot and pruning remains the same (Figure 3-Figure Supplement 2).”

We also added more details to the description of our model in the Methods section:

p.24 line 615, “For all simulations in this study, we distributed nine postsynaptic somata at regular distances in a grid formation on a 2-dimensional 185 × 185 pixel area, representing a cortical sheet (where 1 pixel = 1 micron, Figure 4). This yields a density of around 300 neurons per 𝑚𝑚2 (translating to around 5,000 per 𝑚𝑚3, where for 25 neurons in Figure 3Figure Supplement 2 this would be around 750 neurons per 𝑚𝑚2 or 20,000 per 𝑚𝑚3). The explored densities are a bit lower than compared to neuron densities observed in adulthood (Keller et al., 2018). In the same grid, we randomly distributed 1,500 potential synapses, yielding an initial density of 0.044 potential synapses per 𝜇𝑚2 (Figure 3e). At the end of the simulation time, around 1,000 potential synapses remain, showing that the density of potential synapses is sufficient and does not significantly affect the final density of connected synapses. Thus, the rapid slowing down of growth in our model is not due to a depletion of potential synaptic partners. The resulting density of stably connected synapses is approximately 0.015 synapses per 𝜇𝑚2 (around 60 synapses stabilized per dendritic tree, Figure 3b). This density compares well to experimental findings, where, especially during early development, synaptic densities are described to be within a range similar to the one observed in our model (Leighton et al., 2024; Ultanir et al., 2007; Glynn et al., 2011; Yang et al., 2014; Koshimizu et al., 2009; Tyler and Pozzo-Miller, 2001).”

(III) Lastly, we investigated how the correlation between synapses of the same activity group might affect our conclusions. As correlations in our model mainly arise from patterns of spontaneous activity which are abundant in early postnatal development retinal waves (Ackman et al., 2012) or endogenous activity in the form of highly synchronized events involving a large fraction of the cells (Siegel et al., 2012), we explored varying the correlations within each activity group, across activity groups and combinations of both. While this analysis supported our previously described intuition on how competition between synaptic activities should drive activity-dependent refinement, recently a study found direct evidence for such subcellular refinement of synaptic inputs specifically dependent on spontaneous activity between retinal ganglion cell axons and retinal waves in the superior colliculus (Matsumoto et al., 2024). The new analysis confirmed our earlier results that the competition between activity groups leads to activity-dependent refinement and yielded further insight into how the studied activity correlations can affect the competition. Those results are presented in a completely new figure (new Figure 5, supported by the Supplementary Figure 5-1 and 5-2) and discussed in the Results section:

p.11 line 249, “Group activity correlations shape synaptic overshoot and selectivity competition across synaptic groups.

Since correlations between synapses emerge from correlated patterns of spontaneous activity abundant during postnatal development (Ackman et al., 2012; Siegel et al., 2012), we explored a wide range of within-group correlations in our model (Figure 5a). Although a change in correlations within the group has only a minor effect on the resulting dendritic lengths (Figure 5b) and overall dynamics, it can change the density of connected synapses and thus also affect the number of connected synapses to which each dendrite converges throughout the simulations (Figure 5c,e). This is due to the change in specific selectivity of each dendrite which is a result of the change in within-group correlations (Figure 5d). While it is easier for perfectly correlated activity groups to coexist within one dendrite (Figure 5-Figure Supplement 1a, 100%), decreasing within-group correlations increases the competition between groups, producing dendrites that are selective for one specific activity group (60%, Figure 5d, Figure 5-Figure Supplement 1a). This selectivity for a particular activity group is maximized at intermediate (approximately 60%) within-group correlations, while the contribution of the second most abundant group generally remains just above random chance levels (Figure 5-Figure Supplement 1a). Further reducing within-group correlations (20%, Figure 5a) causes dendrites to lose their selectivity for specific activity groups due to the increased noise in the activity patterns (20%, Figure 5a). Overall, reducing within-group correlations increases synapse pruning (Figure 5f, bottom), also found experimentally (Matsumoto et al., 2024) as dendrites require an extended period to fine-tune connections aligned with their selectivity biases. This phenomenon accounts for the observed reduction in both the density and number of synapses connected to each dendrite.

In addition to the within-group correlations, developmental spontaneous activity patterns can also change correlations between groups as for example retinal waves propagated in different domains (Feller et al., 1997) (Figure 5-Figure Supplement 2). An increase in between-group correlations in our model intuitively decreases competition between the groups since fully correlated global events synchronize the activity of all groups (Figure 5-Figure Supplement 2). The reduction in competition reduces pruning in the model, which can be recovered by combining cross-group correlations with decreased within-group correlations (Figure 5-Figure Supplement 2). Our simulations show that altering the correlations within activity groups increases competition (by lowering the within-group correlations) or decreases competition (by raising the across-group correlations). Hence, in our model, competition between activity groups due to non-trivially structured correlations is necessary to generate realistic dynamics between activity-independent growth and activity-dependent refinement or pruning.

In sum, our simulations demonstrate that our model can operate under various correlations in the spike trains. We find that the level of competition between synaptic groups is crucial for the activity-dependent mechanisms to either potentiate or depress synapses and is fully consistent with recent experimental evidence showing that the correlation between spontaneous activity in retinal ganglion cells axons and retinal waves in the superior colliculus governs branch addition vs. elimination (Matsumoto et al., 2024)."

Precise details on the implementation of the changed activity correlations were added to the Methods section:

p. 25 line 638, “Within-group and across-group activity correlations. For the decreased withingroup correlations, we generated parent spike trains for each individual group with the firing rate 𝑟𝑖𝑛 = 𝑟𝑡𝑜𝑡𝑎𝑙 ∗ 𝑃𝑖𝑛 (e.g., 𝑃𝑖𝑛 = 100%; 60%; 20%, Figure 5). All the synapses of the same group share the same parent spike train and the remaining spikes for each synapse are uniquely generated with the firing rate 𝑟𝑟𝑒𝑠𝑡 = 𝑟𝑡𝑜𝑡𝑎𝑙 ∗ (1 − 𝑃𝑖𝑛) (e.g., (1 − 𝑃𝑖𝑛) = 0%; 40%; 80%), resulting in the desired firing rate 𝑟𝑡𝑜𝑡𝑎𝑙 (see Table 1). For the increase in across-group correlations, we generated one master spike train with the firing rate 𝑟𝑐𝑟𝑜𝑠𝑠 = 𝑟𝑡𝑜𝑡𝑎𝑙 ∗ 𝑃𝑐𝑟𝑜𝑠𝑠 for all the synapses of all groups (e.g., 𝑃𝑐𝑟𝑜𝑠𝑠 = 5%; 10%; 20%, Figure 5-Figure Supplement 2). This master spike train is shared across all groups and then filled up according to the within-group correlation (if not specified differently 𝑃𝑖𝑛 = 1 − 𝑃𝑐𝑟𝑜𝑠𝑠 to maintain the rate 𝑟𝑡𝑜𝑡𝑎𝑙). In all the cases, also in those where the change in across-group correlations is combined with the change in within-group correlations, the remaining spikes for each synapse are generated with a firing rate 𝑟𝑟𝑒𝑠𝑡 = 𝑟𝑡𝑜𝑡𝑎𝑙 ∗ (1 − 𝑃𝑖𝑛 − 𝑃𝑐𝑟𝑜𝑠𝑠) to obtain an overall desired firing rate of 𝑟𝑡𝑜𝑡𝑎𝑙.”

Point 1.2. Many potentially important phenomena seem to be excluded. I realize that no model can be complete, but the choice of which phenomena to include or exclude from this model could bias studies that make use of it and is worth serious discussion. The development of axons is concurrent with dendrite outgrowth, is highly dynamic, and perhaps better understood mechanistically. In this model, the inputs are essentially static. Growing dendrites acquire and lose growth cones that are associated with rapid extension, but these do not seem to be modeled. Postsynaptic firing does not appear to be modeled, which may be critical to activity-dependent plasticity. For example, changes in firing are a potential explanation for the global changes in dendritic pruning that occur following the outgrowth phase.

Thanks to the reviewer for bringing up these important considerations. We do indeed write in the Introduction (e.g. lines 36-76) which phenomena we include in the model and why. The Discussion also compares our model to others (lines 433-490), pointing out that most models either focus on activity-independent or activity-dependent phases. We include both, combining the influence of both molecular gradients and growth factors as well as activity-dependent connectivity refinements instructed by spontaneous activity. We consider our model a tractable, minimalist mechanistic model which includes both activity-independent and activity-dependent aspects.

Regarding postsynaptic firing, this is indeed super relevant and an important point to consider. In one of our recent publications (Kirchner and Gjorgjieva, 2021), we studied only an activity-dependent model for the organization of synaptic inputs on non-growing dendrites which have a fixed length. There, we considered the effect of postsynaptic firing (via a back-propagating action potential) and demonstrated that it plays an important role in establishing a global organization of synapses on the entire dendritic tree of the neuron. For example, we showed that it could lead to the emergence of retinotopic maps on the dendritic tree which have been found experimentally (Iacaruso et al., 2017). Since we use the same activity-dependent plasticity model in this paper, we expect that the somatic firing will have the same effect on establishing synaptic distributions on the entire dendritic tree. This is now also discussed in the Discussion section of the revised manuscript:

p. 21 line 491, “Although we did not explicitly model postsynaptic firing, our previous work with static dendrites has shown that it can play an important role in establishing a global organization of synapses on the entire dendritic tree of the neuron (Kirchner and Gjorgjieva, 2021). For example, we showed that it could lead to the emergence of retinotopic maps on the dendritic tree which have been found experimentally (Iacaruso et al., 2017). Since we use the same activity-dependent plasticity model in this paper, we expect that the somatic firing will have the same effect on establishing synaptic distributions on the entire dendritic tree.”

Including the concurrent development of axons in the model is indeed very interesting. In fact, a recent tour-de-force techniques paper found similar to what we assume. Hebbian activity-dependent dynamics of axonal branches of retinal ganglion cells experiencing spontaneous activity in relation to retinal waves in the superior colliculus (Matsumoto et al., 2024). New branches tend to be added at the locations where spontaneous activity of individual branches is more correlated with retinal waves, whereas asynchronous activity is associated with branch elimination. We suspect the same Hebbian activity-dependent dynamics to apply also to dendritic growth.

To address simultaneous dynamic axons to our growing dendrites, in the revised version of the manuscript, we included a simplified form of axonal dynamics by allowing changes in the lifetime and location of potential synapses, which come from axons of presynaptic partners. We explored different median lifetimes of synapses in combination with several distances with which a synapse can move in the simulated space (new Supplementary Figure 3-3). Our results show that dynamically moving synapses only affect the dynamics and stability of our model when the rate of moving synapses combined with the distance of moving synapses is faster than the dendritic growth. In scenarios in which synapses can move across large distances, dendrites get further destabilized due to synapses transferring from one dendrite to another, perturbing the attractor fields of the potential synapses even in late phases of the simulations. Besides such non-biological scenarios, dynamically moving synapses do not affect the model dynamics too much. Thus, they mostly add additional noise and variability to the growth and pruning without changing the timing and amplitude of the dynamics. These results are discussed in the results section of the revised manuscript:

p.9 line 207, “The development of axons is concurrent with dendritic growth and highly dynamic Matsumoto et al. (2024). To address the impact of simultaneously growing axons, we implemented a simple form of axonal dynamics by allowing changes in the lifetime and location of potential synapses, originating from the axons of presynaptic partners (Figure 3-Figure Supplement 3). When potential synapses can move rapidly (median lifetime of 1.8 hours), the model dynamics are perturbed quite substantially, making it difficult for the dendrites to stabilize completely (Figure 3–Figure Supplement 3c). However, slowly moving potential synapses (median lifetime of 18 hours) still yield comparable results (Figure 3-Figure Supplement 3). The distance of movement significantly influenced results only when potential synaptic lifetimes were short. For extended lifetimes, the moving distance had a minor impact on the dynamics, predominantly affecting the time required for dendrites to stabilize. This was the result of synapses being able to transfer from one dendrite to another, potentially forming new long-lasting connections even at advanced stages of synaptic refinement. In sum, our results show that potential axonal dynamics only affect the stability of our model when these dynamics are much faster than dendritic growth.”

Precise details on the implementation of the dynamically moving synapses and their synaptic lifetimes are now in the Methods section:

p. 25 line 650, “Dynamically moving synapses. For the moving synapses we introduced lifetimes for each synapse, randomly sampled from a log-normal distribution with median 1.8h (for when they move frequently), 4.5h or 18h (for when they move rarely) and variance equal to 1 (Figure 3-Figure Supplement 3b). The lifetime of a synapse decreases only when the synapse is not connected to any of the dendrites (i.e., is a potential synapse). When the lifetime of a synapse expires, the synapse moves to a new location with a new lifetime sampled from the same log-normal distribution. This enables synapses to move multiple times throughout a simulation. The exact locations and distances to which each synapse can move are determined by a binary matrix (dimensions: 𝑝𝑖𝑥𝑒𝑙𝑑𝑖𝑠𝑡𝑎𝑛𝑐𝑒 × 𝑝𝑖𝑥𝑒𝑙𝑑𝑖𝑠𝑡𝑎𝑛𝑐𝑒) representing a ring (annulus) with the inner radius 𝑑/4 and outer radius 𝑑/2 , where the synapse location is at the center of the matrix. All the locations of the matrix within the ring boundaries (between the inner radius and outer radius) are potential locations to which the synapse can move. The synapse then moves randomly to one of the possible locations where no other synapse or dendrite is located. For the movement distances, we chose the ring dimensions 3 × 3, 25 × 25 and 101 × 101, yielding the moving distances (radii) of 1 pixel per movement, 12 pixels per movement and 50 pixels per movement (𝑟 = (𝑑−1)/2). These pixel distances represent small movements, as much as a dendrite can grow in one step (1 micron), and larger movements which are far enough so that the synapse will not attract the same branches again (12 microns) or far enough so that it might attract a completely different dendrite (50 microns, Figure 3-Figure Supplement 3a).”

*Point 1.3. Line 167. There are many ways to include activity -independent and -dependent components into a model and not every such model shows stability. A key feature seems to be that larger arbors result in reduced growth and/or increased retraction, but this could be achieved in many ways (whether activity dependent or not). It's not clear that this result is due to the combination of activity-dependent and independent components in the model, or conceptually why that should be the case.*

We never argued for model uniqueness. There are always going to be many different models (at different spatial and temporal scales, at different levels of abstraction). We can never study all of them and like any modeling study in systems neuroscience we have chosen one model approach and investigated this approach. We do compare the current model to others in the Discussion. If the reviewers have a specific implementation that we should compare our model to as an alternative, we could try, but not if this means doing a completely separate project.

*Point 1.4. Line 183. The explanation of overshoot in terms of the different timescales of synaptic additions versus activity-dependent retractions was not something I had previously encountered and is an interesting proposal. Have these timescales been measured experimentally? To what extent is this a result of fine-tuning of simulation parameters?*

We found that varying the amount of BDNF controls the timescale of the activity-dependent plasticity (see our Figure 6c). Hence, changing the balance between synaptic additions vs. retractions is already explored in Figure 6e and f. Here we show that the overshoot and retraction does not have to be fine-tuned but may be abolished if there is too much activity-dependent plasticity.

Regarding the relative timescales of synaptic additions vs. retractions: since the first is mainly due to activity-independent factors, and the second due to activity-dependent plasticity, the questions is really about the timescales of the latter two. As we write in the Introduction (lines 61-63), manipulating activity-dependent synaptic transmission has been found to not affect morphology but rather the density and specificity of synaptic connections (Ultanir et al. 2007), supporting the sequential model we have (although we do not impose the sequence, as both activity-independent and activitydependent mechanisms are always “on”; but note that activity-dependent plasticity can only operate on synapses that have already formed).

The described results are robust to parameter variations (performed on the postsynaptic density, potential synapse density, and within- and across-group correlations) as described in the reply to reviewer #1 point 1.1.

Point 1.5. Line 203. This result seems at odds with results that show only a very weak bias in the tuning distribution of inputs to strongly tuned cortical neurons (e.g. work by Arthur Konnerth's group). This discrepancy should be discussed.

First, we note that the correlated activity experienced by our modeled synapses (and resulting synaptic organization) does not necessarily correspond to visual orientation, or any stimulus feature, for that matter, but is rather a property of correlated spontaneous activity.

Nonetheless, there is some variability in what the experimental data show. Many studies have shown that synapses on dendrites are organized into functional synaptic clusters: across brain regions, developmental ages and diverse species from rodent to primate (Kleindienst et al., 2011; Takahashi et al., 2012; Winnubst et al., 2015; Gökçe et al., 2016; Wilson et al., 2016; Iacaruso et al., 2017; Scholl et al., 2017; Niculescu et al., 2018; Kerlin et al., 2019; Ju et al., 2020, Hedrick et al., 2022, Hedrick et al., 2024). Interestingly, some *in vivo* studies have reported lack of fine-scale synaptic organization (Varga et al., 2011; X. Chen et al., 2011; T.-W. Chen et al., 2013; Jia et al., 2010; Jia et al., 2014), while others reported clustering for different stimulus features in different species. For example, dendritic branches in the ferret visual cortex exhibit local clustering of orientation selectivity but do not exhibit global organization of inputs according to spatial location and receptive field properties (Wilson et al. 2016; Scholl et al., 2017). In contrast, synaptic inputs in mouse visual cortex do not cluster locally by orientation, but only by receptive field overlap, and exhibit a global retinotopic organization along the proximal-distal axis (Iacaruso et al., 2017). We proposed a theoretical framework to reconcile these data: combining activity-dependent plasticity similar to the BDNF-proBDNF model that we used in the current work, and a receptive field model for the different species (Kirchner and Gjorgjieva, 2021). This is now also discussed in the Discussion section of the revised manuscript:

p. 20 line 471, “The correlated activity experienced by our modeled synapses (and resulting synaptic organization) does not necessarily correspond to visual orientation, or any stimulus feature, for that matter, but is rather a property of spontaneous activity. Nonetheless, there is some variability in what the experimental data show. Many have shown that synapses on dendrites are organized into functional synaptic clusters: across brain regions, developmental ages and diverse species from rodent to primate (Kleindienst et al., 2011; Winnubst et al., 2015; Iacaruso et al., 2017; Scholl et al., 2017; Niculescu et al., 2018; Takahashi et al., 2012; Gökçe et al., 2016; Wilson et al., 2016; Kerlin et al., 2019; Ju et al., 2020; Hedrick et al., 2022, 2024). Other studies have reported lack of fine-scale synaptic organization (Chen et al., 2013; Varga et al., 2011; Chen et al., 2011; Jia et al., 2010, 2014). Interestingly, some of these discrepancies might be explained by different species showing clustering with respect to different stimulus features (orientation or receptive field overlap) (Scholl et al., 2017; Wilson et al., 2016; Iacaruso et al., 2017). Our prior work proposed a theoretical framework to reconcile these data: combining activity-dependent plasticity as we used in the current work, and a receptive field model for the different species (Kirchner and Gjorgjieva, 2021).”

Point 1.6. Line 268. How does the large variability in the size of the simulated arbors relate to the relatively consistent size of arbors of cortical cells of a given cell type? This variability suggests to me that these simulations could be sensitive to small changes in parameters (e.g. to the density or layout of presynapses).

We again thank the reviewer for the detailed explanation and feedback on parameters that should be tested in more detail. We have explored several of the suggested model parameters and believe that we have managed to explain and illustrate their effects on the model's dynamics clearly. The precise changes are explained in the reply to point 1.1 and are now available in the revised version of the manuscript.

Point 1.7. The modeling of dendrites as two-dimensional will likely limit the usefulness of this model. Many phenomena- such as diffusion, random walks, topological properties, etc - fundamentally differ between two and three dimensions.

Indeed, there are many differences between two and three dimensions. We have ongoing work that extends the current model to 3D but is beyond the scope of the current paper. In systems neuroscience, people have found very interesting results making such simplified geometric assumptions about networks, for instance the one-dimensional ring model has been used to uncover fundamental insights about computations even though highly simplified and abstracted. We are convinced that our model, especially with the new sensitivity analysis, makes interesting and novel contributions and predictions.

Point 1.8. The description of wiring lengths as 'approximately optimal' in this text is problematic. The plotted data show that the wiring lengths are several deviations away from optimal, and the random model is not a valid instantiation of the 2D non-overlapping constraints the authors imposed. A more appropriate null should be considered.

We appreciate the reviewer’s feedback regarding the use of the term “approximately optimal” in describing wiring lengths. We acknowledge that our initial terminology was imprecise and could be misleading. We had previously referred to the minimal wiring length as the optimal wiring length, which does not fully capture the nuances of neuronal wiring optimization. As noted in prior literature, such as the work by Hermann Cuntz (Cuntz et al., 2010 & 2012), neurons can optimize their wiring beyond simply minimizing dendritic length.

To address this issue, to better capture the balance between wiring minimization and functional constraints, such as conduction delays, we have developed a new modeling approach based on minimum spanning trees with a balancing factor (Cuntz et al., 2010 & 2012). This factor modulates the trade-off between minimizing wiring length and accounting for conduction delays from synapses to the soma. Specifically, the model assumes a balance between minimizing the total dendritic length and minimizing the tree distance between synapses and the site of input integration, typically the soma. This balance is illustrated in Figure 8 (Figure 7 in the original manuscript), where we demonstrate that the deviation from the theoretical minimum length arises because direct paths to synapses often require longer dendrites in our models.

Together with the new result, which we added as the new panels f, g and h to Figure 8 (originally Figure 7), we also adjusted panel a of Figure 8, to now illustrate the difference between random wiring, minimal wiring and minimal conductance delay. The updated Figure 8 and its new findings are discussed in the results section of the revised manuscript:

p.17 line 387, “This deviation is expected given that real dendrites need to balance their growth processes between minimizing wire while reducing conduction delays. The interplay between these two factors emerges from the need to reduce conduction delays, which requires a direct path length from a given synapse to the soma, consequently increasing the total length of the dendritic cable. (Cuntz et al., 2010, 2012; Ferreira Castro et al., 2020).

To investigate this further, we compared the scaling relations of the final morphologies of our models with other synthetic dendritic morphologies generated using a previously described minimum spanning tree (MST) based model. The MST model balances the minimization of total dendritic length and the minimization of conduction delays between synapses and the soma. This balance results in deviations from the theoretical minimum length because direct paths to synapses often require longer dendrites (Cuntz et al., 2008, 2010). The balance in the model is modulated by a balancing factor (𝑏𝑓). If 𝑏𝑓 is zero, dendritic trees minimize the cable only, and if 𝑏𝑓 is one, they will try to minimize the conduction delays as much as possible. It is important to note that the MST model does not simulate the developmental process of dendritic growth; it is a phenomenological model designed to generate static morphologies that resemble real cells.

To facilitate the comparison of total lengths between our simulated and MST morphologies, we generated MST models under the same initial conditions (synaptic spatial distribution) as our models and simulated them to match several morphometrics (total length, number of terminals, and surface area) of our grown morphologies. This allowed us to create a corresponding MST tree for each of our synthetic trees. Consequently, we could evaluate whether the branching structures of our models were accurately predicted by minimum spanning trees based on optimal wiring constraints. We found that the best match occurred with a trade-off parameter 𝑏𝑓 = 0.9250 (Figure 8f). Using the morphologies generated by the MST model with the specified trade-off parameter (𝑏𝑓), we showed that the square root of the synapse count and the total length (𝐿) in both our model generated trees and the MST trees exhibit a linear scaling relationship (Figure 8g; 𝑅2 = 0.65). The same linear relationship can be observed for the square root of the surface area and the total length 𝐿 of our model trees and the MST trees (Figure 8h; 𝑅2 = 0.73). Overall, these results indicate that our model generate trees are wellfitted by the MST model and follow wire optimization constraints.

We acknowledge that the value of the balancing factor 𝑏𝑓 in our model is higher than the range of balancing factors that is typically observed in the biological dendritic counterparts, which generally ranges between 0.2 and 0.4 (Cuntz et al., 2012; Ferreira Castro et al., 2020; Baltruschat et al., 2020). However, it is still remarkable that our model, which does not explicitly address these two conservation laws, achieves approximately optimal wiring. Why do we observe such a high 𝑏𝑓 value? We reason that two factors may contribute to this. First, in our models, local branches grow directly to the nearest potential synapse, potentially taking longer routes instead of optimally branching to minimize wiring length (Wen and Chklovskii, 2008). Second, the growth process in our models does not explicitly address the tortuosity of the branches, which can increase the total length of the branches used to connect synapses. In the future, it will be interesting to add constraints that take these factors into account. Taken together, combining activity-independent and -dependent dendrite growth produces morphologies that approximate optimal wiring.”

Further details on the fitted MST model and the corresponding analysis were added to the methods section:

p.26 line 669, “Comparison with wiring optimization MST models. To evaluate the wire minimization properties of our model morphologies (n=288), we examined whether the number of connected synapses (N), total length (L), and surface area of the spanning field (S) conformed to the scaling law 𝐿 ≈ 𝜋−1/2 ⋅ 𝑆1/2 ⋅ 𝑁1/2 (Cuntz et al., 2012). Furthermore, to validate that our model dendritic morphologies scale according to optimal wiring principles, we created simplified models of dendritic trees using the MST algorithm with a balancing factor (bf). This balancing factor adjusts between minimizing the total dendritic length and minimizing the tree distance between synapses and the soma (Cost = 𝐿 + 𝑏𝑓 ⋅ 𝑃 𝐿) (MST_tree; best bf = 0.925) (Cuntz et al., 2010); TREES Toolbox http://www.treestoolbox.org.

Initially, we generated MSTs to connect the same distributed synapses as our models. We performed MST simulations that vary the balancing factor between 𝑏𝑓 = 0 and 𝑏𝑓 = 1 in steps of 0.025 while calculating the morphometric agreement by computing the error (Euclidean distance) between the morphologies of our models and those generated by the MST models. The morphometrics used were total length, number of terminals, and surface area occupied by the synthetic morphologies.”

Point 1.9. It's not clear to me what the authors are trying to convey by repeatedly labeling this model as 'mechanistic'. The mechanisms implemented in the model are inspired by biological phenomena, but the implementations have little resemblance to the underlying biophysical mechanisms. Overall my impression is that this is a phenomenological model intended to show under what conditions particular patterns are possible. Line 363, describing another model as computational but not mechanistic, was especially unclear to me in this context.

What we mean by mechanistic is that we implement equations that model specific mechanisms i.e. we have a set of equations that implement the activity-independent attraction to potential synapses (with parameters such as the density of synapses, their spatial influence, etc) and the activitydependent refinement of synapses (with parameters such as the ratio of BDNF and proBDNF to induce potentiation vs depression, the activity-dependent conversion of one factor to the other, etc). This is a bottom-up approach where we combine multiple elements together to get to neuronal growth and synaptic organization. This approach is in stark contrast to the so-called top-down or normative approaches where the method would involve defining an objective function (e.g. minimal dendritic length) which depends on a set of parameters and then applying a gradient descent or other mathematical optimization technique to get at the parameters that optimize the objective function. This latter approach we would not call mechanistic because it involves an abstract objective function (who could say what a neuron or a circuit should be trying to optimize?) and a mathematical technique for how to optimize the function (we don’t know if neurons can compute gradients of abstract objective functions).

Hence our model is mechanistic, but it does operate at a particular level of abstraction/simplification. We don’t model individual ion channels, or biophysics of synaptic plasticity (opening and closing of NMDA channels, accumulation of proteins at synapses, protein synthesis). We do, however, provide a biophysical implementation of the plasticity mechanism through the BDNF/proBDNF model which is more than most models of plasticity achieve, because they typically model a phenomenological STDP or Hebbian rule that just uses activity patterns to potentiate or depress synaptic weights, disregarding how it could be implemented. To the best of our understanding, this is what is normally considered mechanistic in the field (in contrast to, for example, biophysical).

**Reviewer #2 (Public Review):**
This work combines a model of two-dimensional dendritic growth with attraction and stabilisation by synaptic activity. The authors find that constraining growth models with competition for synaptic inputs produces artificial dendrites that match some key features of real neurons both over development and in terms of final structure. In particular, incorporating distance-dependent competition between synapses of the same dendrite naturally produces distinct phases of dendritic growth (overshoot, pruning, and stabilisation) that are observed biologically and leads to local synaptic organisation with functional relevance. The approach is elegant and well-explained, but makes some significant modelling assumptions that might impact the biological relevance of the results.Strengths:The main strength of the work is the general concept of combining morphological models of growth with synaptic plasticity and stabilisation. This is an interesting way to bridge two distinct areas of neuroscience in a manner that leads to findings that could be significant for both. The modelling of both dendritic growth and distance-dependent synaptic competition is carefully done, constrained by reasonable biological mechanisms, and well-described in the text. The paper also links its findings, for example in terms of phases of dendritic growth or final morphological structure, to known data well.Weaknesses:The major weaknesses of the paper are the simplifying modelling assumptions that are likely to have an impact on the results. These assumptions are not discussed in enough detail in the current version of the paper.(1) Axonal dynamics.A major, and lightly acknowledged, assumption of this paper is that potential synapses, which must come from axons, are fixed in space. This is not realistic for many neural systems, as multiple undifferentiated neurites typically grow from the soma before an axon is specified (Polleux & Snider, 2010). Further, axons are also dynamic structures in early development and, at least in some systems, undergo activity-dependent morphological changes too (O'Leary, 1987; Hall 2000). This paper does not consider the implications of joint pre- and post-synaptic growth and stabilisation.

We thank the reviewer for the summary of the strengths and weaknesses of the work. While we feel that including a full model of axonal dynamics is beyond the scope of the current manuscript, some aspects of axonal dynamics can be included and are now implemented and tested in the revised manuscript. Since this feedback covers similar aspects of the model that were also pointed out by reviewer #1, we refer here to our detailed reply to their comments 1.1 and 1.2, where we list and discuss all the analyses performed to address the raised issues.

(2) Activity correlationsOn a related note, the synapses in the manuscript display correlated activity, but there is no relationship between the distance between synapses and their correlation. In reality, nearby synapses are far more likely to share the same axon and so display correlated activity. If the input activity is spatially correlated and synaptic plasticity displays distance-dependent competition in the dendrites, there is likely to be a non-trivial interaction between these two features with a major impact on the organisation of synaptic contacts onto each neuron.

We have explored the amount of correlation (between and within correlated groups) in the revised manuscript (see also our reply to reviewer comment 1.1).

However, previous experimental work, (e.g. Kleindienst et al., 2011) has provided anatomical and functional analyses that it is unlikely that the functional synaptic clustering on dendritic branches is the result of individual axons making more than one synapse (see pg. 1019).

(3) BDNF dynamicsThe models are quite sensitive to the ratio of BDNF to proBDNF (eg Figure 5c). This ratio is also activity-dependent as synaptic activation converts proBDNF into BDNF. The models assume a fixed ratio that is not affected by synaptic activity. There should at least be more justification for this assumption, as there is likely to be a positive feedback relationship between levels of BDNF and synaptic activation.

The reviewer is correct. We used the BDNF-proBDNF model for synaptic plasticity based on our previous work (Kirchner and Gjorgjieva, 2021).

There, we explored only the emergence of functionally clustered synapses on static dendrites which do not grow. In the Methods section (Parameters and data fitting) we justify the choice of the ratio of BDNF to proBDNF from published experimental work. We also performed sensitivity analysis (Supplementary Fig. 1) and perturbation simulations (Supplementary Fig. 3), which showed that the ratio is crucial in regulating the overall amount of potentiation and depression of synaptic efficacy, and therefore has a strong impact on the emergence and maintenance of synaptic organization. Since we already performed all this analysis, we expect that the same results will also apply to the current model which includes dendritic growth, as it involves the same activity-dependent mechanism.

A further weakness is in the discussion of how the final morphologies conform to principles of optimal wiring, which is quite imprecise. 'Optimal wiring' in the sense of dendrites and axons (Cajal, 1895; Chklovskii, 2004; Cuntz et al, 2007, Budd et al, 2010) is not usually synonymous with 'shortest wiring' as implied here. Instead, there is assumed to be a balance between minimising total dendritic length and minimising the tree distance (ie Figure 4c here) between synapses and the site of input integration, typically the soma. The level of this balance gives the deviation from the theoretical minimum length as direct paths to synapses typically require longer dendrites. In the model this is generated by the guidance of dendritic growth directly towards the synaptic targets. The interpretation of the deviation in this results section discussing optimal wiring, with hampered diffusion of signalling molecules, does not seem to be correct.

We agree with this comment. We had wrongly used the term “optimal wiring” as neurons can optimize their wiring not only by minimizing their dendritic length but other factors as noted by the reviewer. In the revised manuscript we replaced the term “optimal wiring” with “minimal wiring” wherever it was incorrectly used. On top of that, we performed further analysis and discussed these differences, as pointed out in the reply to reviewer #1 point 1.8.

To summarize, we want to again thank the reviewer for their in-depth review and all the suggestions that helped us improve the analysis and implementation of our model.

**Reviewer #3 (Public Review):**
The authors propose a mechanistic model of how the interplay between activity-independent growth and an activity-dependent synaptic strengthening/weaken model influences the dendrite shape, complexity and distribution of synapses. The authors focus on a model for stellate cells, which have multiple dendrites emerging from a soma. The activity independent component is provided by a random pool of presynaptic sites that represent potential synapses and that release a diffusible signal that promotes dendritic growth. Then a spontaneous activity pattern with some correlation structure is imposed at those presynaptic sites. The strength of these synapses follow a learning rule previously proposed by the lab: synapses strengthen when there is correlated firing across multiple sites, and synapses weaken if there is uncorrelated firing with the relative strength of these processes controlled by available levels of BDNF/proBDNF. Once a synapse is weakened below a threshold, the dendrite branch at that site retracts and loses its sensitivity to the growth signalThe authors run the simulation and map out how dendrites and synapses evolve and stabilize. They show that dendritic trees growing rapidly and then stabilize by balancing growth and retraction (Figure 2). They also that there is an initial bout of synaptogenesis followed by loss of synapses, reflecting the longer amount of time it takes to weaken a synapse (Figure 3). They analyze how this evolution of dendrites and synapses depends on the correlated firing of synapses (i.e. defined as being in the same "activity group"). They show that in the stabilized phase, synapses that remain connected to a given dendritic branch are likely to be from same activity group (Figure 4). The authors systemically alter the learning rule by changing the available concentration of BDNF, which alters the relative amount of synaptic strengthening, which in turn affects stabilization, density of synapses and interestingly how selective for an activity group one dendrite is (Figure 5). In addition the authors look at how altering the activity-independent factors influences outgrowth (Figure 6). Finally, one of the interesting outcomes is that the resulting dendritic trees represent "optimal wiring" solutions in the sense that dendrites use the shortest distance given the distribution of synapses. They compare this distribute to one published data to see how the model compared to what has been observed experimentally.There are many strengths to this study. The consequence of adding the activity-dependent contribution to models of synapto- and dendritogenesis is novel. There is some exploration of parameters space with the motivation of keeping the parameters as well as the generated outcomes close to anatomical data of real dendrites. The paper is also scholarly in its comparison of this approach to previous generative models. This work represented an important advance to our understanding of how learning rules can contribute to dendrite morphogenesis.

We thank the reviewer for the positive evaluation of the work and the suggestions below.

To improve the clarity of the manuscript, we adjusted and fixed some figures and corresponding paragraphs as follows:

(1) We increased the number of ticks and their corresponding numbers in all the figures to make them easier to read and interpret.

(2) In Figure 3 panel d, showing the evolution of synaptic weight, we corrected the upper limit at the yaxis to 1 (from previously 2).

(3) Due to a typo in the implementation of the BDNF concentration, we had to correct the used BDNF concentrations from 49%, 45% and 40%, to 49%, 46.5% and 43% respectively.

(4) The y-axis labels of Figure 6 (old Figure 5) panel e and f were changed to make the plots clearer (e: “morphology change explained (%)” to "effect on morphology (%)", and f: “synapse connection explained (%)” to "effect on connected synapses (%)").

(5) The values for the eta and tau-w in the supplementary Table were corrected. Previously tau-w was falsely 6000 time steps which was corrected to 3000 time steps, and eta was 45% and is now 46.5%.

We believe that all the changes to the manuscript will address the reviewer’s concerns and enhance the clarity and accuracy of the findings described in the manuscript.